# Osteoclast-mediated resorption primes the skeleton for successful integration during axolotl limb regeneration

**Camilo Riquelme-Guzmán[1], Stephanie L Tsai[2,3], Karen Carreon Paz[1], Congtin Nguyen[1], David Oriola[4,5,6,7†], Maritta Schuez[1], Jan Brugués[4,5,6,7], Joshua D Currie[8], Tatiana Sandoval-Guzmán[9,10]***

[1]CRTD/ Center for Regenerative Therapies TU, Center for Molecular and Cellular Bioengineering, Technische Universität Dresden, Dresden, Germany; [2]Department of Stem Cell and Regenerative Biology, Harvard University, Cambridge, United States; [3]Center for Regenerative Medicine, Massachusetts General Hospital, Harvard Medical School, Boston, United States; [4]Max Planck Institute of Molecular Cell Biology and Genetics, Dresden, Germany; [5]Center for Systems Biology Dresden, Dresden, Germany; [6]Max Planck Institute for the Physics of Complex Systems, Dresden, Germany; [7]Cluster of Excellence Physics of Life, Technische Universität Dresden, Dresden, Germany; [8]Department of Biology, Wake Forest University, Winston-Salem, United States; [9]Department of Internal Medicine 3, Center for Healthy Aging, University Hospital Carl Gustav Carus at the Technische Universität Dresden, Dresden, Germany; [10]Paul Langerhans Institute Dresden of Helmholtz Centre Munich, University Hospital Carl Gustav Carus at the Technische Universität Dresden, Dresden, Germany

**\*For correspondence:**
tatiana.sandoval_guzman@tu-dresden.de

**Present address:** †European Molecular Biology Laboratory, EMBL Barcelona, Barcelona, Spain

**Competing interest:** The authors declare that no competing interests exist.

**Abstract** Early events during axolotl limb regeneration include an immune response and the formation of a wound epithelium. These events are linked to a clearance of damaged tissue prior to blastema formation and regeneration of the missing structures. Here, we report the resorption of calcified skeletal tissue as an active, cell-driven, and highly regulated event. This process, carried out by osteoclasts, is essential for a successful integration of the newly formed skeleton. Indeed, the extent of resorption is directly correlated with the integration efficiency, and treatment with zoledronic acid resulted in osteoclast function inhibition and failed tissue integration. Moreover, we identified the wound epithelium as a regulator of skeletal resorption, likely releasing signals involved in recruitment/differentiation of osteoclasts. Finally, we reported a correlation between resorption and blastema formation, particularly, a coordination of resorption with cartilage condensation. In sum, our results identify resorption as a major event upon amputation, playing a critical role in the overall process of skeletal regeneration.

## Editor's evaluation
Using a well-established and elegant axolotl limb regeneration model and transgenic reporter strains, the study reveals the potential role of osteoclast-mediated resorption in limb regeneration. The findings from the work provide a new understanding of how the skeleton is primed for regeneration.

## Introduction

The axolotl (*Ambystoma mexicanum*) has the ability to regenerate different body structures, including the limbs. Besides making new cells of the right type at the right place, a successful regeneration requires a functional integration of those new cells with the pre-existing tissue, a process that has not been widely studied. In particular, it remains unknown how early processes impact tissue integration.

In general, regeneration progression is marked by different overlapping phases, which lead to the re-establishment of the missing limb (*Sandoval-Guzmán and Currie, 2018*). Two of the most critical events are the formation of the wound epithelium (WE) and the blastema (*Aztekin, 2021*; *Tanaka, 2016*). The WE is formed by migrating keratinocytes, which close the wound in just a few hours (*Hay and Fischman, 1961*; *Repesh and Oberpriller, 1978*). Importantly, the WE is characterized by the absence of a basal lamina, which enhances the diffusion of important factors for regeneration (*Neufeld and Aulthouse, 1986*; *Repesh and Oberpriller, 1978*). Indeed, the WE is a major regulator of the immune response, tissue histolysis (*Tsai et al., 2020*), and blastema proliferation and patterning (*Boilly and Albert, 1990*; *Ghosh et al., 2008*; *Han et al., 2001*). Notably, several works have demonstrated that the WE is required for blastema formation and thus, regeneration (*Mescher, 1976*; *Tassava and Garling, 1979*; *Thornton, 1957*; *Tsai et al., 2020*).

The blastema is a heterogenous pool of progenitor cells arising from the various tissues at the amputation plane (*Kragl et al., 2009*). The connective tissue (CT), a conglomerate of different cell types, is a critical cell source for the blastema, supplying over 40% of the cells within (*Currie et al., 2016*; *Dunis and Namenwirth, 1977*; *Gerber et al., 2018*; *Muneoka et al., 1986*). A particular case is the skeleton: cells embedded in the tissue do not actively participate in regeneration (*Currie et al., 2016*; *McCusker et al., 2016*), instead, CT cells (dermal and periskeletal) rebuild the new skeleton (*Currie et al., 2016*; *Dunis and Namenwirth, 1977*; *McCusker et al., 2016*; *Muneoka et al., 1986*). Although the skeleton represents more than 50% of the exposed surface upon amputation (*Hutchison et al., 2007*), it is unclear the role the embedded cells play in the remodeling and integration of new tissue.

Undoubtedly, the skeletal system is essential for the limb, serving as a physical scaffold and allowing locomotion. Juvenile axolotls present a cartilaginous skeleton composed of chondrocytes and perichondral cells, which progressively ossifies from the time animals reach sexual maturity (*Riquelme-Guzmán et al., 2021*). In contrast, mammalian appendicular skeleton develops by endochondral ossification, a process where a cartilage anlage is replaced by bone (*Kozhemyakina et al., 2015*). To maintain skeletal homeostasis, a key cell type is the osteoclast, a myeloid-derived population, which mediates the degradation of the cartilage matrix prior to bone formation.

Osteoclasts are giant multinucleated cells with a specialized morphology adapted for skeletal resorption (*Cappariello et al., 2014*). Besides their role in homeostasis, osteoclasts are recruited upon bone injuries or trauma. The most studied case is fracture healing (*Einhorn and Gerstenfeld, 2015*) however, in the context of regeneration, osteoclasts are recruited after fin amputation in zebrafish (*Blum and Begemann, 2015*) and digit tip amputation in mouse (*Fernando et al., 2011*). In urodeles, evidence of osteoclast-mediated resorption is scarce (*Fischman and Hay, 1962*; *Tank et al., 1976*). Nevertheless, the presence of myeloid cells triggered by the amputation has been reported (*Debuque et al., 2021*; *Leigh et al., 2018*; *Rodgers et al., 2020*), and the participation of macrophages was shown to be critical. Indeed, ablating macrophages completely halt regeneration (*Godwin et al., 2013*). Similar results were observed in mouse digit tip amputation, and furthermore, a specific osteoclast inhibition resulted in delayed bone resorption, wound closure, and blastema formation (*Simkin et al., 2017*).

Immune cells are involved in the degradation of the extracellular matrix (ECM) in the vicinity of the amputation plane (*Stocum, 2017*), helping the mobilization of progenitor cells (*Thornton, 1938a*; *Thornton, 1938b*). Accordingly, macrophage ablation resulted in a downregulation of matrix metalloproteinases (*Godwin et al., 2013*). Essential for successful regeneration, histolysis is characterized by the release of proteolytic enzymes (*Huang et al., 2021*; *Vinarsky et al., 2005*; *Yang et al., 1999*; *Yang and Bryant, 1994*) and is partially controlled by the WE (*Tsai et al., 2020*; *Vinarsky et al., 2005*).

Limb regeneration is achieved by a complete amalgamation (integration) of the regenerated structures with the mature tissues. Although the regenerated limb is often considered a perfect replica of the pre-existing limb, in the last decade regeneration fidelity has been addressed by a couple of works. For instance, abnormalities were observed in 80% of larvae due to conspecific bites (*Thompson*

*et al., 2014*), and in over 50% of amputated animals (*Bothe et al., 2021*). However, it is still unknown why such phenotypes are observed, and what entails successful versus unsuccessful regeneration. In this regard, regeneration-specific signals in the stump tissue could prime the limb and promote a successful integration. In the newt *Cynops pyrrhogaster*, structural changes in the ECM of the distal humerus can be observed after an elbow joint amputation, demonstrating a correlation between ECM remodeling and proper joint regeneration as well as integration to the mature tissue (*Tsutsumi et al., 2015*).

With all the aforementioned evidence, we sought to assess the significance of skeletal histolysis for regeneration. We observed a rapid skeletal resorption which is carried out by osteoclasts, and we provide evidence that this process is essential for tissue integration. Moreover, we propose a role for the WE in resorption induction and a spatiotemporal coordination between resorption and blastema formation. Overall, our work provides an in-depth assessment of how a remodeling process influences the final outcome of regeneration using the axolotl limb.

## Results

### Skeletal elements are resorbed upon amputation

To determine the changes in the skeleton upon amputation, we used the stable calcium-binding dyes calcein and alizarin red. These dyes label mineralized cartilage in juvenile axolotls, allowing in vivo imaging (*Riquelme-Guzmán et al., 2021*). Using 4–6 cm snout-to-tail (ST) axolotls, we amputated the zeugopod at the distal end of the calcified tissue and imaged at different days post amputation (dpa) (*Figure 1A*). We observed a consistent reduction in the calcein$^+$ tissue from 7 to 12 dpa. We quantified the length of the calcified tissue in both zeugopodial elements and compared them to the initial length at day 0 (*Figure 1B*). Resorption initiated after 7 dpa and by 12 dpa, over 40% of the calcified radius and 60% of the calcified ulna were resorbed (length resorbed radius: 342.83±95.75 µm; ulna: 770.67±94.34 µm). We pooled five independent experiments and noticed an important variability between assays (*Figure 1C*, each color represents an assay). The median for radius resorption is 40% and for ulna 60%; however, in several cases the calcified tissue was completely resorbed in both elements. Although an inter-assay variability was observed, intra-assay animals presented a consistent resorption ratio.

Digits are a simplified platform to perform in vivo imaging, therefore we assessed resorption by amputating the distal end of the calcein$^+$ tissue in the distal phalanx of the second digit (*Figure 1D*). Similar to the zeugopod, we quantified the calcein$^+$ tissue length at different dpa and revealed a similar trend in the resorptive dynamics: resorption starting after 7 dpa and receding by 13 dpa (*Figure 1E*), vanishing over 50% of the calcified tissue length (320.43±113.56 µm).

Finally, we collected limbs at 9 and 15 dpa and stained them with alcian blue (*Figure 1F*). At 9 dpa, we observed resorption in both radius and ulna. Remarkably, we occasionally observed a break in the ulna (*Figure 1F* arrowhead) that sometimes led to the extrusion of the skeletal fragment through the epidermis. This skeletal shedding was observed both in digit and limb amputations. At 15 dpa, resorption was finished, and the condensation of the new skeleton could be observed. In sum, we report resorption to be a process that occurs upon amputation of different calcified skeletal elements in the axolotl limb.

### Osteoclasts are identified during skeletal resorption

Osteoclasts are specialized multinucleated cells responsible for skeletal resorption (*Charles and Aliprantis, 2014*). Despite their critical role in skeletal biology, osteoclasts have only been reported on the basis of morphology during salamander regeneration (*Fischman and Hay, 1962*; *Nguyen et al., 2017*; *Tank et al., 1976*). Therefore, we sought to identify osteoclasts during resorption using various molecular markers.

Cathepsin K (CTSK) and the tartrate-resistant acid phosphatase (TRAP) are released by osteoclasts and are used as identifying markers (*Cappariello et al., 2014*). Using sections from zeugopodial amputations, we performed immunofluorescence (IF) using an anti-CTSK antibody (*Figure 2A*) and TRAP enzymatic staining (*Figure 2B*). CTSK$^+$ cells were identified in sections at 8 dpa adjacent or inside the calcein$^+$ skeleton. Similarly, TRAP$^+$ cells were identified at 9 dpa. Next, to correlate osteoclast recruitment with resorption timing, we performed RT-qPCR at different dpa using specific

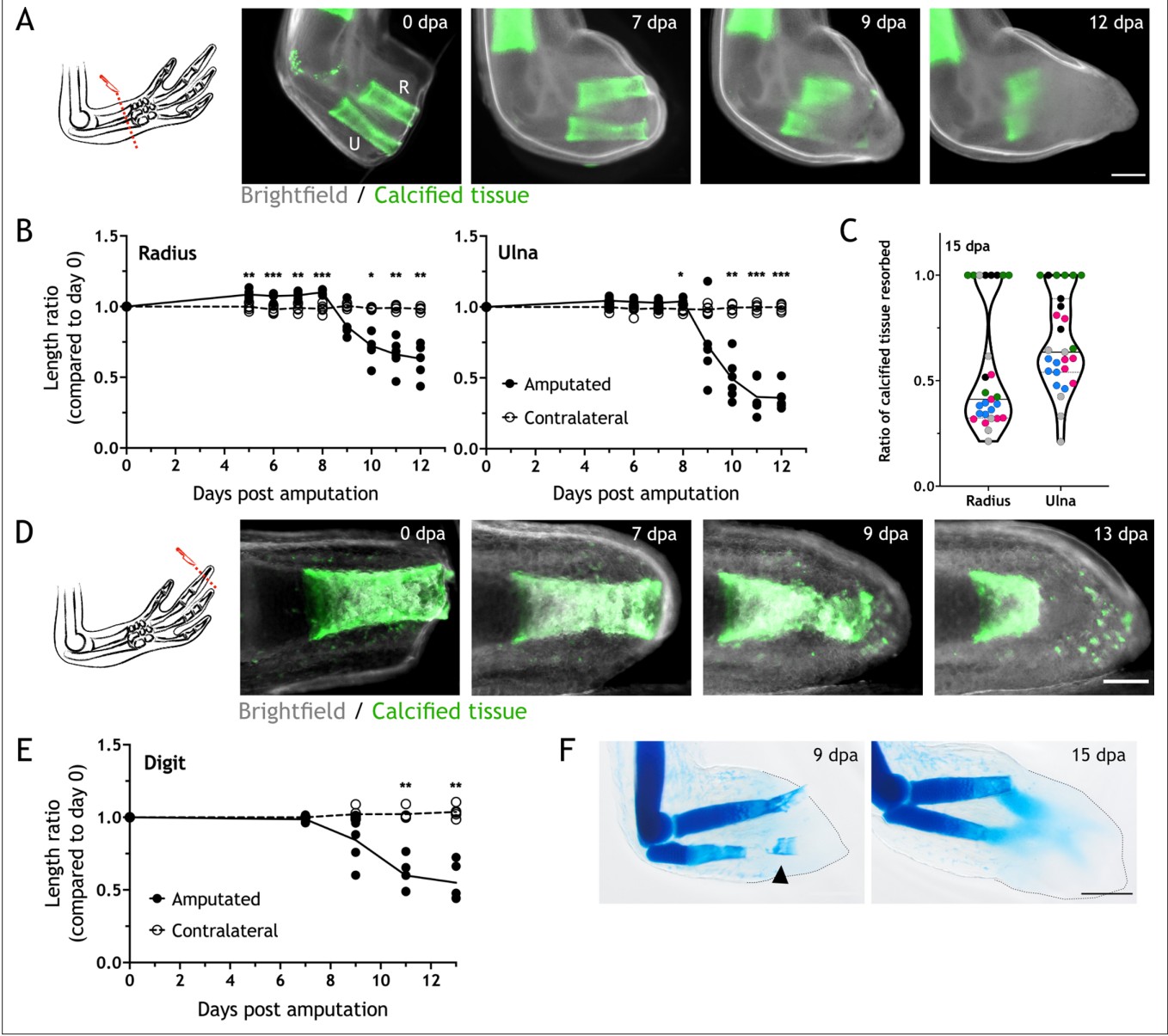

**Figure 1.** Skeletal elements are resorbed upon amputation. (**A**) Time course of resorption during zeugopod regeneration. Calcein-stained axolotls were amputated at the distal end of the calcified tissue. *R*: radius; U: ulna; Scale bar: 500µm. (**B**) Quantification of resorption rate in radius and ulna in (**A**). Length ratio was calculated using the length at 0 dpa as a reference. Each dot represents an animal (n=6; *** p<0.001, ** p<0.01, * p<0.05, two-way ANOVA, Bonferroni's multiple comparisons test, amputated versus contralateral). (**C**) Quantification of resorption percentage in calcified radius and ulna among animals in different assays. Each assay is represented by a color (pool of five independent experiments, n=27). (**D**) Time course of resorption during digit regeneration. Calcein-stained axolotls were amputated at the distal end of the calcified tissue. Scale bar: 200µm. (**E**) Quantification of calcified digit resorption in (**D**). Length ratio was calculated using the length at 0 dpa as a reference. Each dot represents an animal (n=5; ** p<0.01, two-way ANOVA, Bonferroni's multiple comparisons test, amputated versus contralateral). (**F**) Alcian blue staining of limbs at different dpa. Arrowhead: broken piece of ulna. Dashed line: outline of distal limb. Scale bar: 500µm (n=2).

primers for *Trap*, *Ctsk*, and *Dcstamp* (dendritic cell-specific transmembrane protein, involved in osteoclast multinucleation). The RNA relative content for the three markers behaved similarly: a sharp increase was observed, reaching a peak at 9 dpa before rapidly decrease to almost basal levels at 15 dpa (*Figure 2C*).

To assess osteoclast spatiotemporal dynamics in vivo, we generated the *Ctsk:mRuby3* and *Ctsk:eGFP* transgenic lines, which express the fluorescent protein *mRuby3* or *enhanced GFP (eGFP)* under the control of *Ctsk* promoter from zebrafish. In the *Ctsk:eGFP* line, we observed a considerable

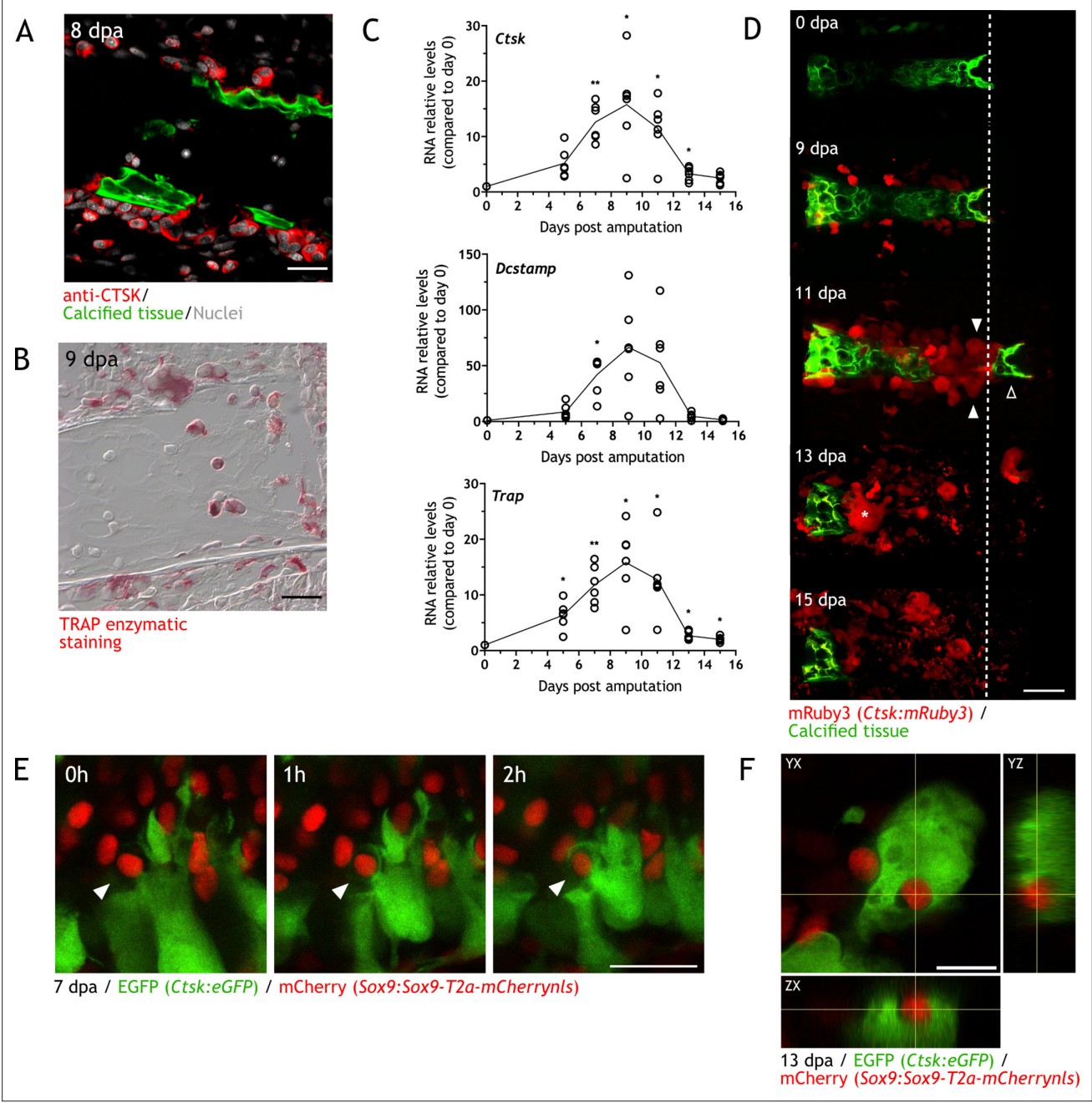

**Figure 2.** Osteoclasts are identified during skeletal resorption. (**A**) Apotome image of immunofluorescence for anti-cathepsin K (CTSK; red) in zeugopod section at 8 dpa. Calcein was used for calcified cartilage labeling (green) and Hoechst for nuclear staining (white). Scale bar: 50 μm (n=2). (**B**) Tartrate-resistant acid phosphatase (TRAP) enzymatic staining in zeugopod section at 9 dpa. Scale bar: 50 μm (n=2). (**C**) RT-qPCR for *Trap*, *Ctsk*, and *Dcstamp* at different dpa upon zeugopodial amputation. Solid line represents mean, each dot is an animal (n=6. ** p<0.01, * p<0.05, one-way ANOVA, Bonferroni's multiple comparisons test, each timepoint versus 0 dpa). (**D**) In vivo confocal imaging of *Ctsk:mRuby3* (red) upon digit amputation. Calcein was used for calcified cartilage labeling (green). Image represents a maximum intensity projection of 10 images (3 μm interval). White arrowhead: mRuby3+ cells (osteoclasts). Black arrowhead: break in the skeletal tissue. Dashed line: amputation plane. Asterisk: multinucleated osteoclast. Scale bar: 100 μm (n=3). (**E**) In vivo confocal imaging of *Sox9 × Ctsk* at 7 dpa. Images were taking at time 0, 1, and 2hr. Image represents a maximum intensity projection of four images (3μm interval). White arrowhead: eGFP+ osteoclast engulfing mCherry+ chondrocyte. Scale bar: 50 μm (n=6). (**F**) Orthogonal view of in vivo confocal imaging from the *Sox9 × Ctsk* line at 13 dpa. Image is composed of 15 planes with a voxel depth of 2 μm. Center of cross shows mCherry+ cell phagocytosed by eGFP+ cell. Scale bar: 50 μm (n=6).

The online version of this article includes the following figure supplement(s) for figure 2:

**Figure supplement 1.** Characterization of *Ctsk:eGFP* transgenic line and assessment of apoptosis in skeleton during regeneration.

number of eGFP[+] cells labeled by the F4/80 antibody (*Figure 2—figure supplement 1A*), which generally identifies macrophages (*Gordon et al., 2011*; *Yun et al., 2015*). A previous work identified *Ctsk* as a periosteal stem cell population marker in mice (*Debnath et al., 2018*). In axolotl, we observed a low number of eGFP[+]/F4/80 cells along the periskeleton (*Figure 2* and *Figure 2—figure supplement 1B*), suggesting a similar population in axolotls. With this experiment, we confirm that the bulk of eGFP[+] cells in the vicinity of the skeleton are of monocytic origin and differentially distinguished from periskeletal cells by morphology and F4/80 expression.

Using our *Ctsk:mRuby3* line, we followed resorption in vivo in digits with confocal microscopy (*Figure 2D*, two independent experiments). At 0 dpa, the tissue was devoid of mRuby3[+] cells. At 9 dpa, mononuclear-like mRuby3 [+] cells were observed in the periphery of the calcified phalanx. These cells increased in numbers and became multinucleated at 11 dpa (white arrowheads). A piece of the calcified structure (black arrowhead) has been detached at this timepoint. At 13 dpa, most of the phalanx was resorbed, and mRuby3[+] cells were scattered throughout the sample. A giant multinucleated cell was observed next to the calcified tissue (asterisk *Figure 2D*). Finally, between 13 and 15 dpa, resorption was completed, and mRuby3[+] cells vacated the space. By morphology, some of these cells showed signs of apoptotic puncta and with TUNEL staining, several apoptotic osteoclasts in zeugopod sections were revealed at 11 dpa (*Figure 2—figure supplement 1C*).

During regeneration, apoptosis has been reported shortly after amputation in the WE, muscle, and periosteum (*Bucan et al., 2018*). We evaluated if significant chondrocyte apoptosis could precede resorption, and observed, at both 5 and 7 dpa, few apoptotic cells inside the cartilage (*Figure 2—figure supplement 1D*). However, these apoptotic cells are mainly at the surface facing the amputation site and not throughout the calcified skeletal element to be resorbed. To further evaluate the fate of chondrocytes inside the skeleton, we used a *Sox9 × Ctsk* transgenic line (*Sox9:Sox9-T2a-mCherrynls x Ctsk:eGFP*) and assess whether *Ctsk*[+] cells phagocytose chondrocytes while resorbing the skeletal matrix. The *Sox9:Sox9-T2a-mCherrynls* line labels chondrocytes with the mCherry protein (*Riquelme-Guzmán et al., 2021*). We imaged several engulfing events in which mCherry[+] cells were being surrounded by eGFP[+] cells (*Figure 2E*), and mCherry[+] cells inside eGFP[+] cells (*Figure 2F*). These results provide evidence that osteoclasts phagocytose chondrocytes during resorption. In sum, utilizing different approaches, we demonstrated the presence and substantial participation of osteoclasts in the regeneration-induced resorption.

## Zoledronic acid treatment inhibits osteoclast-mediated skeletal resorption

To assess the effect of osteoclast inhibition on regeneration, we treated animals with the osteoclast inhibitor zoledronic acid (zol). Zol is a potent bisphosphonate, used in the treatment of osteoporosis. It is internalized by osteoclasts, preventing protein prenylation and consequently their intracellular localization and function (*Dhillon, 2016*), which could lead to apoptosis (*Clézardin, 2013*). By serial intraperitoneal injections of 200 µg/kg of zol every 3 days, we evaluated the effect of osteoclast inhibition by imaging the length of the skeletal elements at different dpa. Zol treatment inhibited resorption as seen at 12 dpa (*Figure 3A*, three independent experiments), since most of the calcified tissue remained intact when compared to the untreated control and vehicle. Quantification of both radius and ulna lengths at different dpa revealed a significant difference between the radius or ulna in zol-treated animals compared to the controls at 11, 12, and 15 dpa (*Figure 3B*).

Furthermore, we measured the relative RNA content of *Ctsk*, *Trap*, and *Dcstamp* at 9 dpa in each condition. No significant difference was observed for the three markers (*Figure 3C*), although the mean for zol-treated samples was smaller in each case. Our results suggested that zol treatment mainly results in a consistent inhibition of osteoclast function. Consequently, we performed in vivo imaging of digit regeneration upon zol treatment in the *Ctsk:eGFP* transgenic line. When resorption was inhibited by zol treatment, we observed a reduction in the number of eGFP[+] cells (*Figure 3D*, two independent experiments). Although present, these cells did not seem to resorb the calcified tissue. Therefore, zol treatment inhibits osteoclast-mediated resorption, but it does not result in their complete ablation.

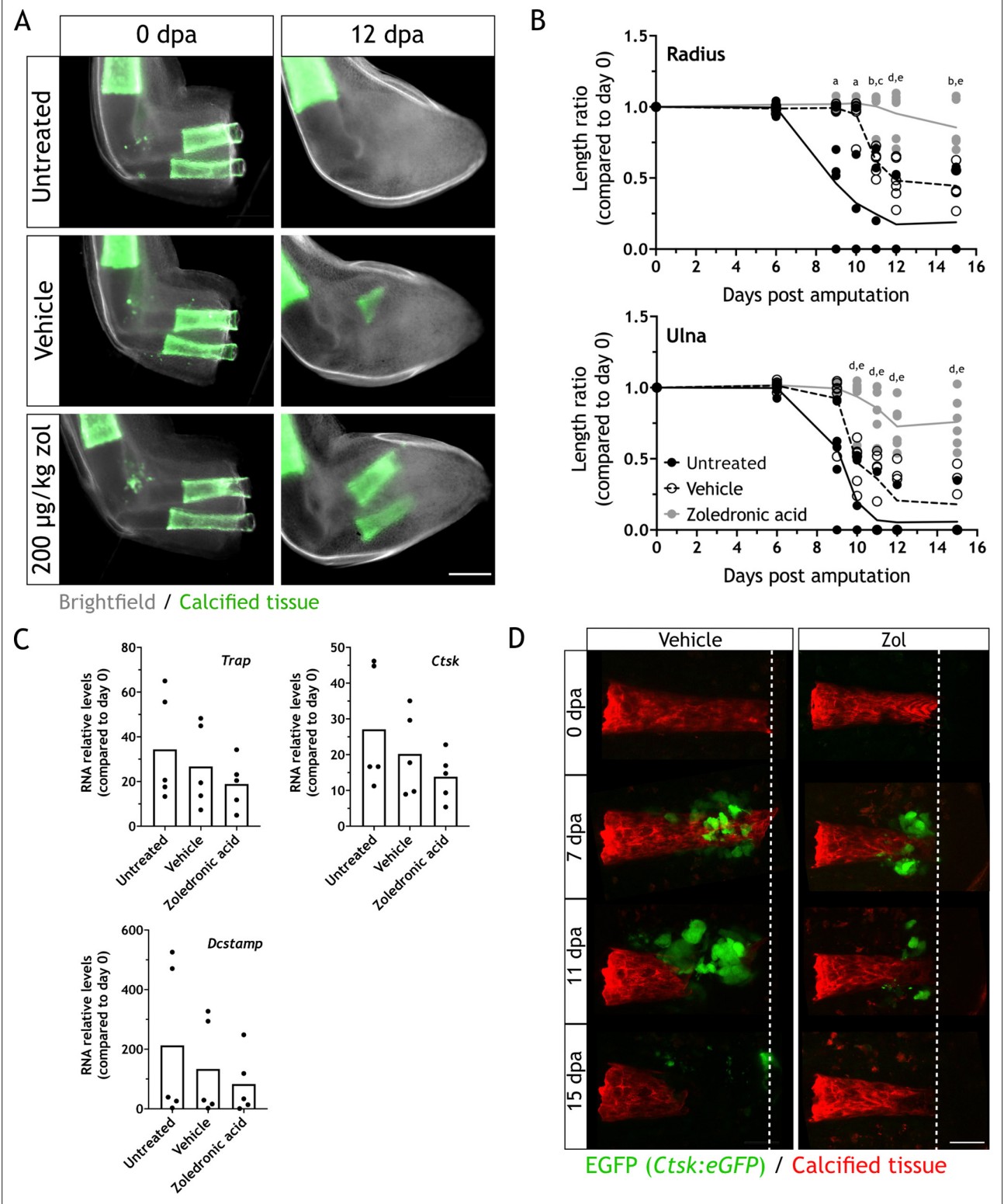

**Figure 3.** Zoledronic acid (zol) treatment inhibits osteoclast-mediated skeletal resorption. (**A**) Time course of resorption during zeugopod regeneration upon zol treatment. Calcein-stained axolotls were amputated at the distal end of the calcified tissue. Scale bar: 500µm. (**B**) Quantification of resorption rate in radius and ulna upon zol treatment in (**A**). Length ratio was calculated using the length at 0 dpa as a reference. Each dot represents an animal (n=6; a: p<0.05 uninjected vs. zol, b: p<0.01 uninjected vs. zol, c: p<0.001 vehicle vs. zol, d: p<0.001 uninjected vs. zol, e: p<0.01 vehicle vs. zol, two-way ANOVA, Tukey's multiple comparisons test). (**C**) RT-qPCR for *Trap*, *Ctsk*, and D*cstamp* at 9 dpa upon zol treatment. Each dot represents an animal (n=5, one-way ANOVA, Tukey's multiple comparisons test). (**D**) In vivo confocal imaging of *Ctsk:eGFP* (green) upon digit amputation. Alizarin red was used for calcified cartilage labeling (red). Image represents a maximum intensity projection of 15 images (3µm interval). Scale bar: 50 µm (n=4).

## Skeletal resorption is necessary for a successful integration of the regenerated structure

To assess the importance of resorption, we followed the zol-treated animals until 45 dpa. At this stage, limbs are fully formed but they have not reached yet full size (*Tank et al., 1976*). Looking at the gross morphology, resorption inhibition did not halt regeneration, as zol-treated animals were able to form a new limb. We assessed integration by sequential stainings using calcium binding dyes of different colors. We distinguished the original calcification (calcein$^+$) from the calcification of regenerated skeletal elements (alizarin red$^+$) (*Figure 4A*, left panel). In contralateral limbs, alizarin red staining showed new calcification after 0 dpa. Comparatively, we observed no calcein$^+$ tissue in the untreated animals, indicating a full resorption of the calcified tissue in the radius and ulna. The alizarin red$^+$ region demonstrated regeneration of the skeleton. In zol-treated animals, at least half of the calcified region was calcein$^+$/alizarin red$^+$, confirming resorption inhibition. Interestingly, we observed a faulty morphology in radii from the zol-treated animals (arrowhead *Figure 4A*, left panel). To gain a better insight into the morphology of the regenerated zeugopod, we collected those limbs and stained them with alcian blue (*Figure 4A*, right panel). The zol-treated limb showed a clear failure in the integration of the newly formed cartilage, especially in the radius. The new tissue lacked a seamless connection to the stump, presenting an angulated morphology (black arrowhead). In the ulna, heterotopic cartilage formation was seen (asterisk). Surprisingly, the skeletal elements of untreated regenerating animals also showed imperfect morphology, even though the calcified areas were fully resorbed. Both radius and ulna were restored as one complete unit, but with an irregular interphase between the stump and regenerated tissue, observed as a narrowing in the mid-diaphysis (black arrowhead, *Figure 4A*).

Among the untreated and zol-treated limbs, we found different rates of resorption (*Figure 4B*). A correlation between resorption rate and integration efficiency could be observed, particularly for the radius. In a zol-treated animal, with null resorption in the radius (R:0), the distal end of the stump and the proximal end of the regenerated skeleton failed to meet. The regenerated skeleton formed at an adjacent plane, therefore lacking continuity with the pre-existing skeleton. Moreover, a secondary condensation zone was seen, as cartilage also formed distal to the unresorbed tissue. To consistently quantify integration success, we generated a score matrix (Materials and methods), which provided values of 0–4 to the different integration phenotypes. Faithfully regenerated elements had lower values than malformed elements. We calculated the score for each radius and ulna and correlated them with the resorption rate (*Figure 4C*, *Figure 4—figure supplement 1*). For both skeletal elements there was a correlation between resorption and integration success, particularly for the radius, in which the most dramatic phenotypes were observed. In the ulna, we report less severe phenotypes (scores 0–3), but still a significant correlation between increased malformations and decreased resorption rate.

Next, we assessed whether the faulty integration was resolved at later stages. We collected untreated limbs at 90 dpa and stained them with alcian blue/alizarin red. In 6/6 limbs screened, in which the resorption rate was over 50% for both elements, we could still observe a faulty integration of both radius and ulna (arrowheads, *Figure 4D*). This imperfect integration was identified by an angulation at the stump-regenerated interphase, similar to what we reported at 45 dpa.

As resorption has a clear impact on skeletal integration during regeneration, we sought to analyze the ECM organization and its changes at the stump-regenerated interphase using quantitative polarization microscopy (LC-PolScope) (*Oldenbourg, 1996*). In zol-treated limbs, the phenotype is disruptive and thus hinders any attempt to assess ECM organization, therefore, we used limb sections from normally regenerated animals for our analysis (i.e. untreated). By looking at the ECM organization (retardance image), we observed a clear difference in the hypertrophic zone (HZ) of regenerated ulnas when compared to the contralateral limb (arrowhead, *Figure 4E*). We believe that this region in the HZ corresponded to the stump-regenerated interphase. Next, using the retardance image, we created a digital mask that allowed us to quantify the orientation of the ECM components using the slow axis orientation image (*Figure 4E*, lower panels). We defined two regions, the HZ, where the interphase is found, and the resting zone (RZ), a control region proximal to the amputation plane. We generated a histogram representing the angle distribution in each zone. We observed that the HZ in the contralateral ulnas presented a parallel organization of the ECM respect to the proximodistal axis, while the regenerated presented a shift in the organization, with the ECM fibers arranged perpendicularly. The RZ remained unchanged in both sample sets (*Figure 4F*). This result shows that the regenerated ECM

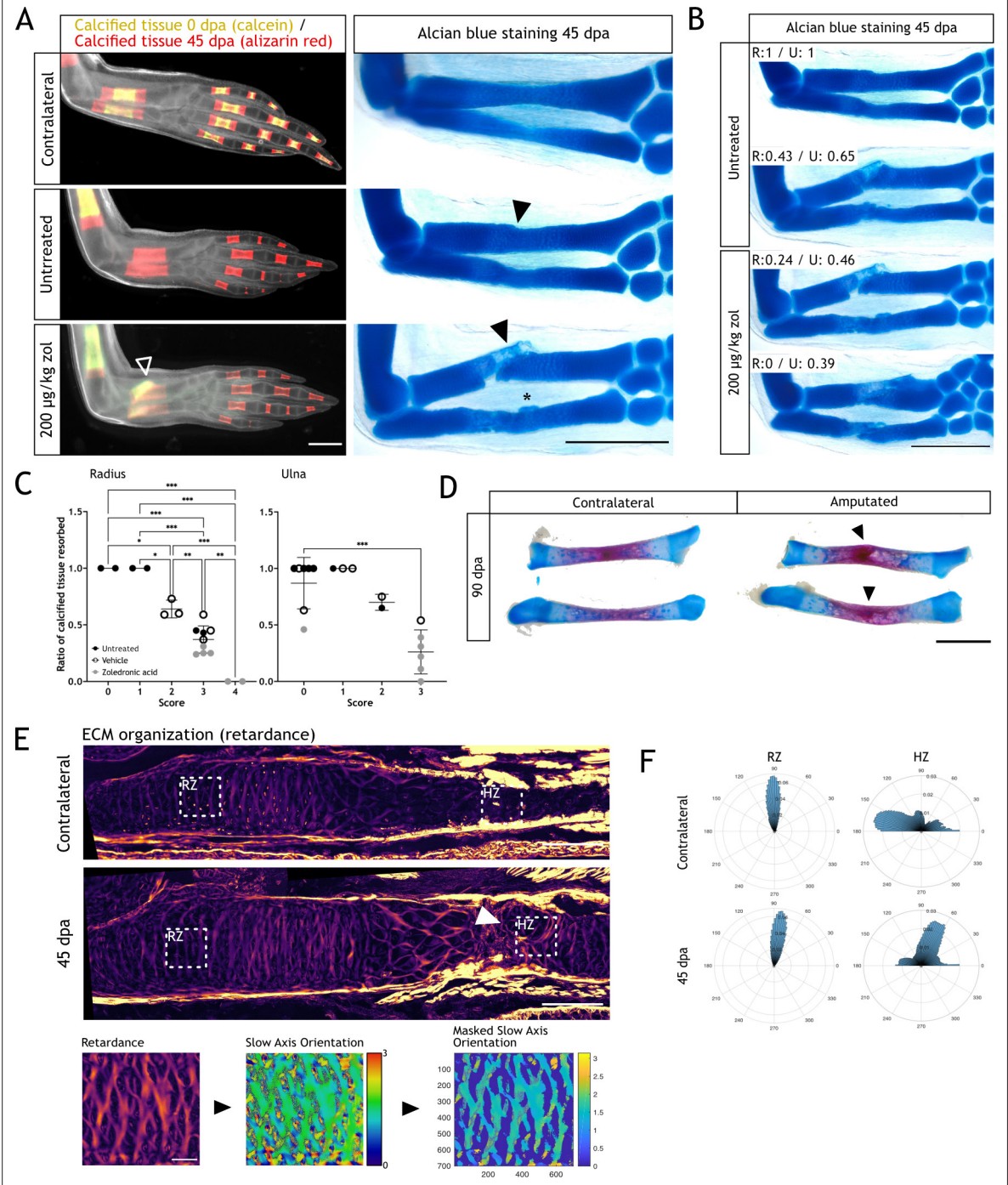

**Figure 4.** Resorption inhibition does not halt regeneration but results in an integration failure of the newly formed skeleton. (**A**) In vivo calcein/alizarin red staining (left panel) and alcian blue staining (right panel) in zoledronic acid (zol)-treated limbs at 45 dpa. Arrowheads: integration failure in skeletal elements. Asterisk: heterotopic cartilage formation in ulna. Scale bar: 1 mm (n=6). (**B**) Alcian blue staining in zol-treated limbs at 45 dpa. Resorption rate for radius and ulna is specified for each case. Scale bar: 1 mm (n=6). (**C**) Correlation between integration scores (Materials and methods) and resorption rate for radius (left) and ulna (right). Each dot represents an animal (n=18. * p<0.05, ** p<0.01, *** p<0.001, one-way ANOVA, Tukey's multiple comparisons test). (**D**) Alcian blue/alizarin red staining of zeugopodial elements at 90 dpa. Arrowhead: stump-regenerated interphase. Scale bar: 2 mm. (**E**) Upper panel: retardance image from unamputated and 40 dpa ulna. RZ (resting zone) and HZ (hypertrophic zone) squares represent the quantification areas. Arrowhead: disorganized interphase. Scale bar: 200μm. Lower panel: quantification flow chart. The mask was created using the retardance image to quantify only extracellular matrix (ECM) components and applied to the slow axis orientation image to determine the orientation of the ECM components at each pixel. In the masked orientational field, the cellular regions are shown in dark blue for visualization purposes but their orientational values were excluded from the analysis in F. Scale bar: 50 μm (n=7 for unamputated, n=9 for amputated). (**F**) Histograms showing the

*Figure 4 continued*
orientation of the ECM components at each pixel in RZ or HZ for the unamputated or 40 dpa ulna. Angles are shown in degrees (n=7 for unamputated, n=9 for amputated).

The online version of this article includes the following source data and figure supplement(s) for figure 4:

**Source data 1.** Score matrix for integration phenotypes.

**Figure supplement 1.** Limbs used for scoring integration phenotypes.

does not recapitulate the original structure and supports the idea that skeletal regeneration is not completely efficient in the axolotl.

Altogether, these results show the importance of resorption during skeletal regeneration and its requirement for integration of the regenerated tissue. Furthermore, these results also highlight that even in normal conditions, the regenerated skeleton does not recapitulate the smooth structure seen pre-amputation.

## The WE is involved in resorption induction

A previous report showed that the WE is critical for inflammation and tissue histolysis (***Tsai et al., 2020***). When the WE formation was prevented by mechanically closing the wound with stump tissue, in a so-called full skin flap (FSF) surgery, *Ctsk* expression was absent at 5 dpa compared to control. This suggests potential defects in skeletal resorption and a role of the WE in its induction. Therefore, we sought to evaluate the role of the WE in skeletal resorption.

Given the technical difficulty of this surgical procedure, we used 14 cm ST animals, similar to the previous reported model (***Tsai et al., 2020***). We amputated the limbs prior to FSF surgery, and followed them until 15 dpa. Using in vivo imaging, we observed an inhibition in resorption in FSF samples by comparing calcified tissue length to the control limb (arrowhead ***Figure 5A***). Next, we collected the limbs and performed alcian blue/alizarin red staining. In seven out of nine control samples, we observed a clear degradation in the distal end of the skeletal elements (black arrowhead ***Figure 5B***), while no or limited resorption was observed in the FSF limbs. To further confirm resorption inhibition, we collected limbs at 9 dpa and performed in situ hybridization (ISH) for *Ctsk*. We saw a significant reduction of the *Ctsk* staining in FSF sections (***Figure 5C***), suggesting the WE plays a role in the recruitment or differentiation of osteoclasts.

By using 14 cm ST animals, we demonstrate that resorption also occurs when skeletal elements in the limb are undergoing ossification. Limbs in older animals are opaquer and become harder to image to quantify the length of the calcified tissue, thus, we performed µCT scans in limbs of animals 16 cm ST (***Figure 5—figure supplement 1***). We confirmed a significant resorption of ossified elements in a slightly extended, but conserved time window as in small animals.

To evaluate whether the WE position might determine the region of resorption initiation, we spatially correlated resorption and the WE. We performed whole mount ISH (WISH) for *Krt17*, which labels cells in the basal layers of the WE (***Leigh et al., 2018***). We observed a clear labeling of the WE from 1 to 7 dpa (***Figure 5D***). In all cases, at least one-third of the skeletal elements (yellow dashed lines) were covered by the WE, which could account for over 50% of the tissue that will be resorbed. Morphologically, the WE is characterized by the absence of a basal lamina (***Neufeld and Aulthouse, 1986***; ***Repesh and Oberpriller, 1978***; ***Tsai et al., 2020***); hence, we used this feature to correlate the WE and resorption in untreated animals. We collected and sectioned limbs at 1, 5, and 7 dpa and performed Masson's trichrome staining. The lack of a basal lamina was observed by the absence of collagen staining in blue (yellow arrowheads, ***Figure 5E***). Osteoclasts could be identified by their multiple nuclei and morphology. As expected, we did not observe any osteoclast at 1 dpa. At 5 dpa, we identified the first infiltration of the skeletal elements, albeit a small number of osteoclasts (white arrowheads, ***Figure 5E and F***). Finally, we saw pronounced infiltration at 7 dpa, including the presence of osteoclasts (white arrowheads, ***Figure 5E and G***). Notably, most of these cells were located in the proximity of the WE. To evaluate the location of osteoclasts, we defined a region by drawing a line between the edges of the WE, and we mapped the position of each osteoclast at 7dpa (***Figure 5H***). Most of the osteoclasts were located in the region covered by the WE, i.e., in the distal part of the skeletal elements. In the more proximal regions of the zeugopod, we did not observe any osteoclasts. This analysis was confirmed using our *Ctsk:eGFP* line at 7 dpa. We imaged sections for eGFP and

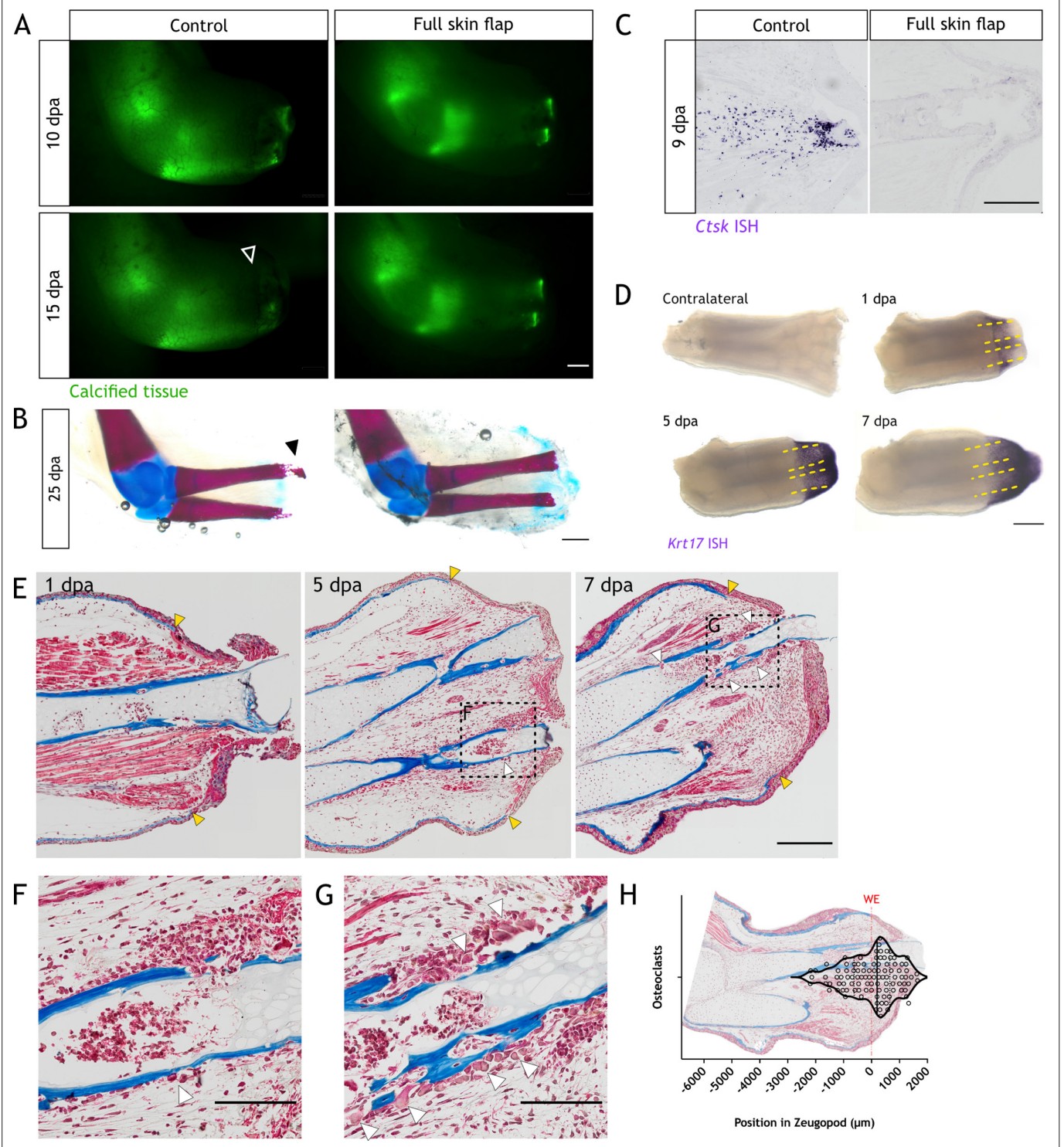

**Figure 5.** The wound epithelium is involved in resorption induction. (**A**) Time course of resorption during zeugopod regeneration upon full skin flap (FSF) surgery. Calcein-stained axolotls were amputated at the distal end of the calcified tissue. Arrowheads: resorption in control cases. Scale bar: 1 mm (n=9). (**B**) Alcian blue/alizarin red staining of limbs at 25 dpa after FSF surgery. Arrowhead: resorption of distal radius. Scale bar: 1 mm (n=9). (**C**) In situ hybridization (ISH) for *Ctsk* in limb sections at 9 dpa after FSF surgery. Scale bar: 500 µm (n=3 for control, n=4 for FSF). (**D**) Whole mount ISH (WISH) for *Krt17* in limbs upon zeugopod amputation at different dpa. Dashed lines: skeletal elements position. Scale bar: 500µm (n=3). (**E**) Masson's trichrome staining from limb sections upon zeugopod amputation at different dpa. Yellow arrowheads: beginning of wound epithelium. White arrowheads: osteoclasts (n=3). (**F**) Inset from (**E**) 5 dpa. Scale bar: 200µm. (**G**) Inset from (**E**) 7dpa. White arrowheads: osteoclasts. Scale bar: 200µm. (**H**) Quantification

*Figure 5 continued on next page*

*Figure 5 continued*

of position of osteoclasts in zeugopod at 7 dpa. Each dot represents an osteoclast. Position of wound epithelium (WE) is shown with a red line. Image of a quantified section shows the position of osteoclasts in the sample (three independent experiments, n=101).

The online version of this article includes the following figure supplement(s) for figure 5:

**Figure supplement 1.** Bones are resorbed upon amputation in 16cm snout-to-tail axolotls.

**Figure supplement 2.** *Ctsk⁺* cells are located in the vicinity of the wound epithelium (WE) at 7 dpa.

nuclear labeling prior to performing Masson's trichrome staining (*Figure 5—figure supplement 2*). Similar to the abovementioned results, we observed eGFP⁺ cells in the proximity of the WE, being most of them next to the skeletal elements.

In sum, our data suggest that the WE plays a role in both osteoclasts recruitment and/or differentiation, and influences the site of resorption initiation in the distal regions of the skeletal elements.

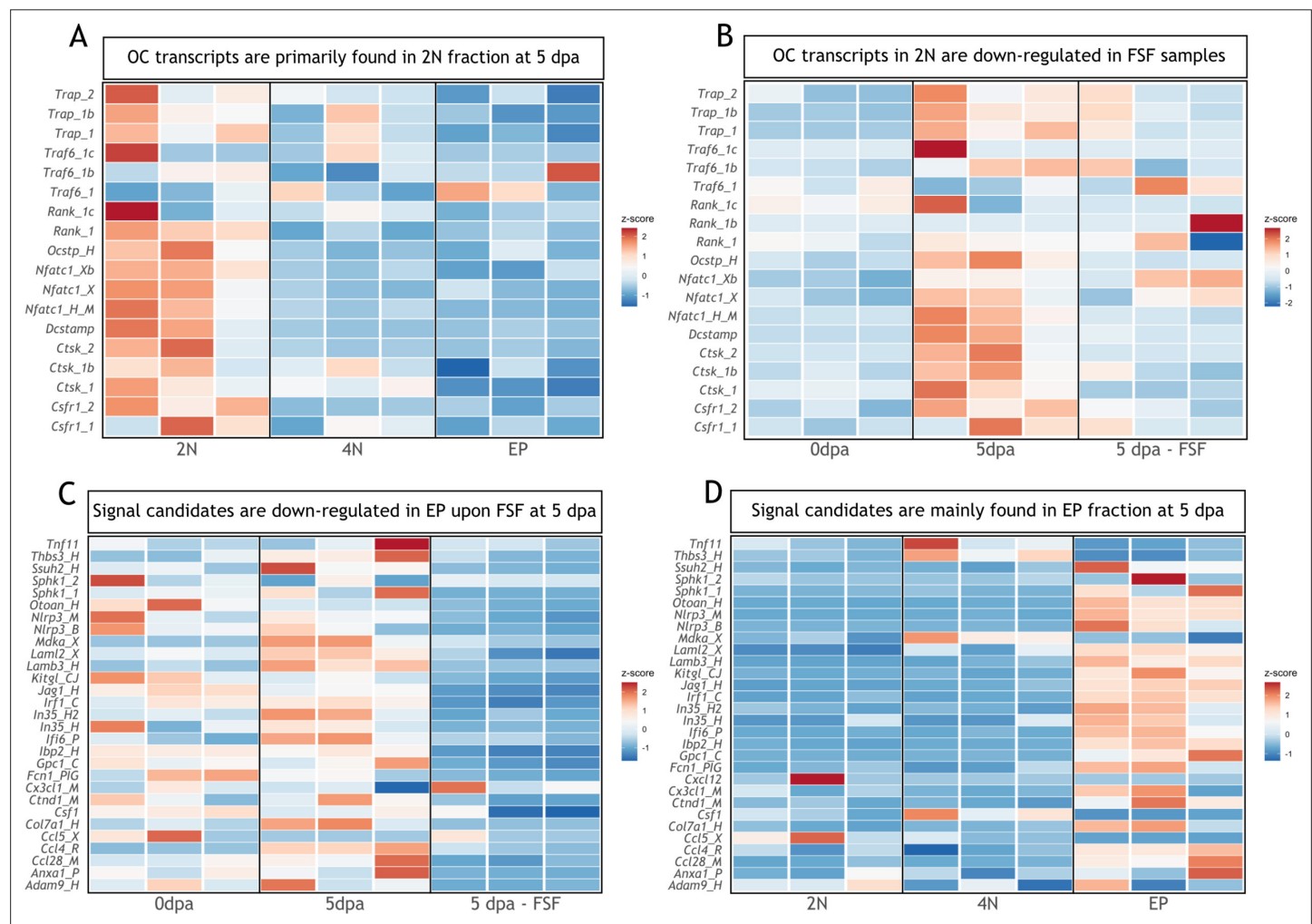

**Figure 6.** Transcripts associated with osteoclastogenesis are downregulated in full skin flap (FSF) samples at 5 dpa. (**A**) Heatmap of transcripts associated with osteoclast function in three different fractions at 5 dpa. 2N: mature cells, 4N: dividing cells, EP: epithelial cells (n=3). (**B**) Heatmap of transcripts associated with osteoclast function in 2N fraction at different timepoints. FSF samples correspond at 5 dpa (n=3). (**C**) Heatmap of differentially downregulated transcripts in EP fraction after FSF surgery at 5 dpa associated with osteoclast recruitment and/or differentiation (n=3). (**D**) Heatmap of differentially downregulated transcripts after FSF surgery in three different fractions associated with osteoclast recruitment and/or differentiation (n=3).

## Identification of candidates involved in osteoclast recruitment and/or differentiation

To identify possible candidates involved in osteoclast recruitment and/or differentiation, we curated a published RNA-seq dataset in which FSF surgery was performed (*Tsai et al., 2020*). In that work, three different populations from blastemas at 5 dpa were isolated: dividing cells (4N), non-dividing cells (2N), and epithelial cells (EP). We first checked which fraction was enriched for transcripts associated with osteoclast function at 5 dpa. As expected, non-dividing cells (2N) were enriched for osteoclast genes (*Trap, Traf6, Rank, Ocstp, Nfatc1, Dcstamp, Ctsk, Csfr1*; *Figure 6A*). Moreover, in the 2N fraction at 5 dpa, most of those transcripts were upregulated compared to day 0, and downregulated in FSF limbs at 5 dpa (*Figure 6B*). This analysis supports our previous results, in which we observed an inhibition of resorption when the WE formation was prevented.

Next, we evaluated which transcripts were significantly downregulated in the EP fraction of FSF limbs compared to a control limb at 5 dpa (supplementary information *Tsai et al., 2020*). We found several transcripts associated with osteoclast recruitment and/or differentiation (*Figure 6C*). From these candidates, previous works report a role for *Ccl4* (*Xuan et al., 2017*), *Sphk1* (*Baker et al., 2010*; *Ishii et al., 2009*; *Ryu et al., 2006*), and *Mdka* (*Maruyama et al., 2004*) in osteoclastogenesis. Moreover, these three transcripts were upregulated at 5 dpa compared to 0 dpa (*Figure 6C*). We confirmed that the candidate transcripts shown in *Figure 6C* were mostly expressed in the EP fraction (*Figure 6D*), including *Sphk1* and *Ccl4. Mdka* levels were more prominent in the 4N fraction; however, it was recently shown that it plays a critical role in WE development and inflammation control during the earlier stages of regeneration (*Tsai et al., 2020*). Our analysis suggests factors expressed in the epithelial fraction as potential candidates regulating osteoclastogenesis during regeneration.

## Skeletal resorption and blastema formation are spatially and temporally correlated

Blastema formation is the accumulation of progenitor cells at the amputation plane. However, taking resorption into consideration, those cells might initially accumulate (or reprogram) more proximal to the amputation plane. Here, we showed that resorption can reach up to 100% of the calcified tissue; and hence, the accumulation of progenitor cells might occur up to 1 mm behind the amputation plane. With this in mind, we sought to assess the blastema specification in the context of skeletal resorption.

First, we measured the blastema surface in images taken at 15 dpa, when resorption is completed and blastema already formed in zol-treated animals. We considered the distal end of the skeletal elements as the starting point of blastema, as it was proposed as the zone where progenitor cells accumulate (*Tank et al., 1976*). As shown in *Figure 7A*, we found a significant decrease in blastema area in zol-treated animals (yellow dashed line). However, the number of proliferative cells (as measured by EdU staining) did not differ (*Figure 7—figure supplement 1A, B*). Importantly, we showed that resorption inhibition does not halt regeneration; and hence, the accumulation of progenitor cells cannot be defined by the position of the skeleton. To efficiently analyze the blastema position during resorption, we used molecular markers. First, we performed whole mount EdU staining at different dpa (*Figure 7B*, two independent experiments). Similar to previous reports, in an intact limb, EdU$^+$ cells are less than 0.5% of the total cells (*Johnson et al., 2018*). At 7 dpa, we observed EdU$^+$ cells behind the amputation plane, spanning over more than 500 μm. These cells were located where we expected to observe resorption. Interestingly, several EdU$^+$ cells were located in the periskeleton (arrowheads), which could account for cells contributing to skeletal regeneration (*Currie et al., 2016*; *McCusker et al., 2016*). These cells were found along most of the skeletal element length. At 10 dpa, when resorption is occurring, we observed a more defined blastema (white arrowhead), which contained EdU$^+$ cells. Similar to 7 dpa, a significant proportion of those EdU$^+$ cells were located next to the resorbing skeleton. Finally, at 15 dpa, we observed an evident reduction in the skeletal length (yellow arrowheads) and a defined blastema distal to those skeletal elements (white dashed line). At this point, very few EdU$^+$ cells were found next to the skeleton.

Although proliferation is mainly observed where blastema is forming, distal migration of EdU$^+$ cells after division is also expected. The site of blastema formation was further assessed at 7 dpa using *Kazald1* as a marker, since it was shown to play a critical role in blastema formation (*Bryant et al., 2017*). Similar to the EdU$^+$ labeling, we observed *Kazald1$^+$* cells located behind the amputation plane, surrounding the distal ends of the skeletal elements (dashed lines), in a zone where resorption is very

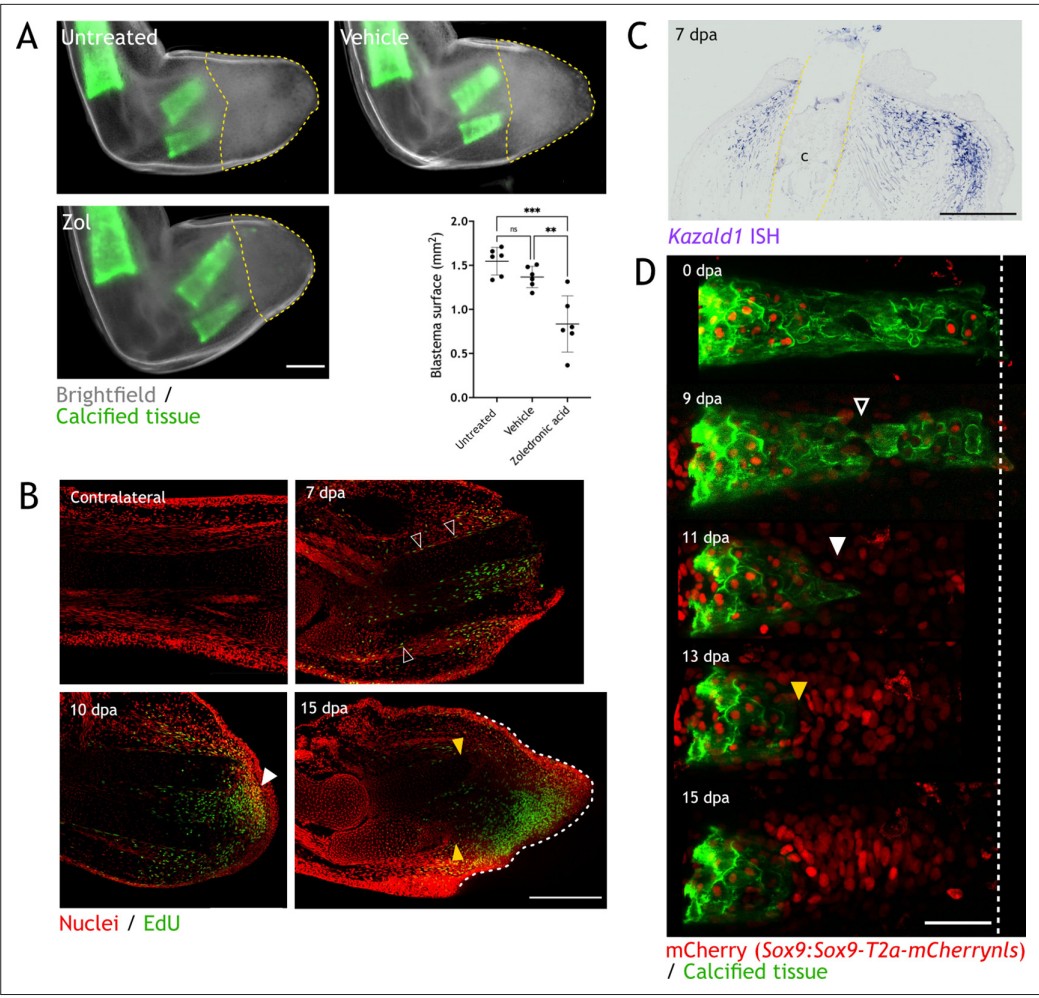

**Figure 7.** Skeletal resorption and blastema formation are spatially and temporally correlated. (**A**) Quantification of blastema size in zoledronic acid-treated limbs at 15 dpa. Dashed lines: blastema. Each dot represents an animal, mean and SD are also shown (n=6, *** p<0.001, ** p<0.01, one-way ANOVA, Tukey's multiple comparisons test). (**B**) Whole mount EdU staining (green) in limbs upon zeugopod amputation at different dpa. TO-PRO-3 was used for nuclear staining (red). Black arrowheads: dividing periskeletal cells. White arrowhead: blastema. Yellow arrowheads: distal end of skeletal element. Dashed line: blastema. Scale bar: 500µm (n=6). (**C**) In situ hybridization (ISH) for *Kazald1* in limb sections upon zeugopod amputation at 7 dpa. Yellow dashed lines: cartilage (c) position. Scale bar: 500 µm (n=3). (**D**) Time course of resorption during digit regeneration in *Sox9-mCherry* (red). Calcein-stained (green) axolotls were amputated at the distal end of the calcified tissue. Black arrowhead: calcified tissue break. White arrowhead: condensation of mCherry⁺ cells. Yellow arrowhead: resorption. Scale bar: 100µm (n=6).

The online version of this article includes the following figure supplement(s) for figure 7:

**Figure supplement 1.** Skeletal resorption and condensation are spatially and temporally correlated.

likely to occur (*Figure 7C*). These results suggest that blastema forms in the same region and at the same time as resorption, and that using the skeletal element as a boundary for blastema identity may provide an incomplete view of the course of regeneration.

Finally, to assess the cellular dynamics of skeletal resorption/formation, we used the *Sox9:Sox9-T2a-mCherrynls* transgenic line in conjunction with calcein staining to follow both processes in vivo. We performed amputation of the distal phalanx and followed the same amputated digit at different dpa (*Figure 7D*). At 0 dpa, no mCherry⁺ cells were found outside the calcified tissue. At 9 dpa, we observed a break in the calcified phalanx (arrowhead), which shows the place where resorption was initiated. At 11 dpa, some mCherry⁺ cells started to group distal to the resorbed calcified tissue (white arrowhead). Those cells represent the initial condensation of the regenerating cartilage. Interestingly, at 13 dpa, resorption continued (yellow arrowhead) and the number and density of mCherry⁺ cells

increased. Finally, resorption was finished at 15 dpa, and the condensation of mCherry[+] cells in the new phalanx presented a defined pattern. The condensation observed from 11 dpa occurred behind the amputation plane and parallel to resorption. Indeed, we observed condensation of mCherry[+] cells while eGFP[+] osteoclasts were still present at the same plane in the skeleton at 11 and 13 dpa (*Figure 7—figure supplement 1C*). In sum, resorption and skeletal regeneration are overlapping processes, as condensation of cartilage progenitors occurs in the presence of osteoclast behind the amputation plane.

## Discussion

Axolotl limb regeneration is an intricate multi-step process that requires the fine-tuning of events such as wound closure, tissue repair, progenitor cells recruitment, and the re-establishment of the functional unit. Although extensive work has been done to understand the cellular dynamics of blastema formation, other processes such as tissue histolysis, the immune response, and tissue integration are yet to be fully understood.

Here, we report that upon digit and zeugopod amputation, significant skeletal resorption is carried out by an osteoclast population and can result in the resorption of 100% of the calcified matrix. Skeletal resorption is observed in young animals with cartilaginous limbs as well as young adult animals with ossified bones. Upon inhibition of resorption, we observed a failure in the integration of the regenerated zeugopodial elements and, interestingly, this failure can also occur in untreated animals undergoing regeneration. Moreover, we present evidence supporting the role of the WE in resorption induction. Finally, we observed a spatial correlation between resorption progression and blastema formation. Particularly, we found that the condensation of new cartilage started before the resorptive process is completed.

Similar to salamanders, mouse digit tip regeneration progresses with early histolysis and blastema formation. An extensive resorption of the phalanx can lead to bone volume reduction of almost 50% of its original size (*Fernando et al., 2011*). When osteoclasts are inhibited, regeneration is not compromised (*Simkin et al., 2017*), and when wound closure is induced earlier, resorption is delayed but regeneration progresses (*Simkin et al., 2015*). This suggests that in mouse digit tip, resorption facilitates blastema formation. Moreover, it has been shown that both periosteal and endosteal cells are responsible for regeneration of the phalanx upon amputation in mice (*Dawson et al., 2018*), suggesting that bone resorption may be required for mobilization of a pool of progenitor cells. Although bone resorption is not required for regeneration to progress, this histolytic process is indeed an important event for the efficient regeneration of the mouse digit tip. In contrast to mouse, axolotl skeletal elements do not mobilize progenitors to the blastema, and wound closure occurs within hours upon amputation, even when skeletal structures protrude at the surface. An important similarity to our findings in axolotl is the time frame when resorption occurs (7–15 dpa). This suggests that the unique fast activation and clearance of osteoclasts are particular to regeneration.

### Resorption efficiency defines skeletal integration success

In this work, osteoclast inhibition with zol resulted in a clear failure in tissue integration. This phenotype often presented as an angulation of the radius, heterotopic cartilage formation in the ulna, or a complete separation between the mature and the regenerated structures. In general, we observed a higher rate of resorption for the ulna than for the radius, which could account for the more striking integration phenotypes reported for the radius.

Our experiments revealed a gradient of integration phenotypes correlated with the amount of tissue resorbed: the more resorption, the better the integration. Strikingly, in animals undergoing normal regeneration we often observed faulty integration phenotypes in mineralized skeleton, as seen by angulations at the stump-regenerated interphase. A recent report showed abnormally regenerated limbs in almost 50% of the animals screened, and they presented similar phenotypes to the ones presented here, i.e., a narrowing in the diaphysis and some heterotopic cartilage formation (*Bothe et al., 2021*). Using polarization microscopy, we presented evidence of an ECM disorganization in the interphase region where both stump and regenerated tissue are connected. Finally, we showed a prevalence of these phenotypes at 90 dpa, proving that they are not resolved after regeneration has been completed.

Remarkably, we observed a high variability in the amount of calcified tissue resorbed in different animals, ranging from 25 to 100% for radius and ulna, being the inter-assay variability higher than the intra-assay. The source of this variability could be an environmental factor (e.g. water temperature), but it highlights the different outcomes that skeletal regeneration can produce. Indeed, in some cases, resorption involved a sequential degradation of the skeletal tissue, and at other times, a break and shedding of a skeleton piece, which has been observed in mouse digit tip (*Fernando et al., 2011*). In contrast to mouse, skeletal shedding in the axolotl is not associated with wound re-epithelialization, since this occurs after wound closure.

Our results highlight the misconception that axolotl limb regeneration recapitulates the pre-existing morphology with high fidelity. This report reveals that a faulty skeletal regeneration is a rather common outcome in the axolotl limb and correlates resorption efficiency with successful skeletal integration.

## WE, resorption, and blastema

The WE is a critical structure for the progression of regeneration, regulating process such as ECM remodeling, tissue histolysis, proliferation, and inflammation (*Tsai et al., 2020*). Here, by blocking the formation of the WE, we underscore its role in the initiation of resorption. Moreover, we analyzed available WE RNA-seq data and found several factors known to influence osteoclast progenitor migration and/or differentiation. Among those candidates, *Sphk1*, *Ccl4,* and *Mdka* are upregulated in the epithelial fraction during regeneration and have been linked to osteoclast biology. Indeed, S1P, which is phosphorylated by the sphingosine kinase, has been shown to have a role in both bone resorption and bone formation (*Ishii et al., 2009*; *Pederson et al., 2008*; *Ryu et al., 2006*), while CCL4 and MDKA have been connected with osteoclast progenitor recruitment (*Maruyama et al., 2004*; *Xuan et al., 2017*). Future studies to understand how these factors are regulated in osteoclast-mediated resorption during regeneration will be needed.

We hypothesized that the connection between the WE and skeletal resorption could be mediated by the WE position. Our results show that resorption starts distally, enclosed by the WE boundaries, supporting this idea. Although the WE may not be the only source of factors inducing osteoclast progenitor migration and differentiation, it probably is a general source of chemokines inducing the recruitment of receptor activator of nuclear factor κ B (RANK)$^+$ myeloid progenitors. We hypothesize that those myeloid cells will then recognize factors secreted by the skeletal elements that promote osteoclast differentiation. Indeed, the main source of RANK ligand (RANKL) in mammals is both hypertrophic chondrocytes and osteocytes (*Xiong et al., 2011*).

In this work, we also provide evidence supporting a spatiotemporal correlation between skeletal resorption and blastema formation. The WE produces signals involved in blastema proliferation and patterning (*Boilly and Albert, 1990*; *Ghosh et al., 2008*; *Han et al., 2001*; *Tsai et al., 2020*), and thus both, resorptive and proliferating signals, could be acting in the same zone of influence. Moreover, since the skeletal elements are structural supports of the limb, the resorption of the hard matrix might cause a collapse of the soft tissue toward the proximal region, and thus a formation of the blastema behind the amputation plane. Indeed, we observed EdU$^+$ cells and *Kazald1$^+$* cells in the surroundings of the skeletal elements shortly before the start of resorption, and condensation of skeletal progenitors distal to the resorbed tissue and under the amputation plane.

Historically, the amputation plane has been conceptualized as a fixed position in the limb which determines the beginning of the blastema. This could be derived from the common practice to trim the skeletal elements right after amputation, because it ensures a consistent blastema formation between experiments. In this surgical procedure, the skin of the amputated limb is retracted, and the extending skeletal elements are re-amputated. Trimming results in a faster WE and blastema formation; however, this procedure might cause the erroneous perception of a fixed amputation plane. Comparatively, in the case of the mouse digit tip, in which bone resorption occurs, a regeneration plane has been identified as more proximal than the amputation plane (*Seifert and Muneoka, 2018*). This study has implications for demarcating the blastema, the progenitor cell source, and the dynamic interphase created by nascent, migrating cells and a stream of morphogens.

## Future perspectives and challenges

There are still unresolved questions regarding the osteoclast population here described. First, what is the origin of this population during regeneration? Upon amputation, a peak of myeloid chemotactic molecules was reported at 1 dpa, followed by an infiltration of myeloid cells and macrophages (*Godwin et al., 2013*). This suggests that osteoclast progenitors are recruited to the amputation plane, but it does not rule out the participation of resident progenitors in the neighboring tissues. Interestingly, axolotl osteoclasts might be different to the more extensively studied mouse osteoclasts. We report here a considerable number of *Ctsk*+ cells to be F4/80+. In contrast, in mouse bone marrow-derived macrophages, F4/80 inhibits osteoclast differentiation (*Kang et al., 2017*). Second, how homogenous is this *Ctsk*+ population? We observed in situ, a range of mononucleated and multinucleated cells resorbing the skeleton. We speculate that as a consequence of resorption speed, some immune cells may skip fusion; and hence, mononucleated osteoclasts are observed. Nevertheless, these speculations need to be addressed in future works. Third, what is the fate of osteoclasts after resorption? We showed that this population eventually vacates the regenerating structure, but before this occurs, some osteoclasts turn positive in an apoptosis staining. Whether all cells undergo apoptosis or if some will recirculate, need further investigation. Recent works have shown that osteoclasts can be long lived (*Jacome-Galarza et al., 2019*) and also be recycled via a cell type named osteomorphs (*McDonald et al., 2021*). These studies underline the need to continue investigating osteoclast biology in vivo, and in general, the rapidly triggered immune response. The axolotl limb presents a new paradigm in which osteoclast function can be assessed, and thus the development of new transgenic lines to label myeloid progenitors and to indelibly label osteoclast will provide a mean to resolve the aforementioned questions. These tools will also allow to specifically assess the function of individual immune cell types, which is lacking when using chemicals as inhibitors. For example, macrophages exhibit some sensitivity to zol (*Mönkkönen et al., 2007*; *Rogers et al., 2013*).

In addition, the concomitant resorption and regeneration need to be further explored. It is known that histolysis helps with the mobilization of progenitor cells in salamanders (*Thornton, 1938b*) and in mouse (*Dawson et al., 2018*), but how osteoclast-mediated resorption could be influencing cartilage condensation in the context of axolotl regeneration remains to be studied. Specifically, how the cell differentiation and migration are orchestrated with respect to resorption is unclear. Of particular interest are periskeletal cells migrating toward the blastema and contributing to the formation of the proximal skeleton. The cell source zone, i.e., a zone where blastema cells are recruited, has been roughly defined to be 500 µm from the amputation plane; however, resorption was not taken into consideration for that assessment (*Currie et al., 2016*). It is unclear how resorption and the detachment and migration of periskeletal cells are coordinated, or if the source of periskeletal cells corresponds to a region not resorbed (e.g. proximal to the calcified tissue). Previous works have demonstrated the interaction between osteoclast and osteoblasts in vivo (*Furuya et al., 2018*; *Ikebuchi et al., 2018*), and the in vivo assessment of this in the context of skeletal regeneration would be necessary.

Finally, we need to consider the location where resorption is occurring since different cell types are found in the same skeletal element along the proximodistal axis, which could influence the outcome of resorption in skeletal regeneration (*Riquelme-Guzmán et al., 2021*).

## Concluding remarks

This work presents a systematic assessment of the timing, extent, and consequences of skeletal resorption. We show that the skeleton undergoes a massive and rapid histolytic event, which is essential for a successful integration of the regenerated structure. This process, which is carried out by osteoclasts, is influenced by the formation of the WE and is correlated with the spatial position of the early blastema. Furthermore, we present proof that the axolotl limb regeneration is not perfect, and it often leads to abnormal skeletal phenotypes. We consider that resorption is playing a key role in skeletal regeneration, and its implications need to be further explored, particularly its coordination with cell migration and condensation of the new skeleton.

# Materials and methods

**Key resources table**

| Reagent type (species) or resource | Designation | Source or reference | Identifiers | Additional information |
|---|---|---|---|---|
| Strain, strain background (*Ambystoma mexicanum*) | Axolotl, white (d/d) | CRTD Axolotl facility | | |
| Genetic reagent (*A. mexicanum*) | *Sox9:Sox9-T2a-mCherrynls* | *Riquelme-Guzmán et al., 2021* | C-Ti^(t/+)(*Sox9:Sox9-T2a-mCherrynls*)^ETNKA | |
| Genetic reagent (*A. mexicanum*) | *Ctsk:mRuby3* | This paper | tgTol2(*Drer.Ctsk:mRuby3*)^tsg | Generated in d/d strain |
| Genetic reagent (*A. mexicanum*) | *Ctsk:eGFP* | This paper | tgTol2(*Drer.Ctsk:eGFP*)^tsg | Generated in d/d strain |
| Genetic reagent (*A. mexicanum*) | *Sox9 × Ctsk* | This paper | C-Ti^(t/+)(*Sox9:Sox9-T2a-mCherrynls*)^ETNKA × tgTol2(*Drer.Ctsk:eGFP*)^tsg | |
| Antibody | Rabbit polyclonal anti-CTSK | Abcam | Cat #ab1902 RRID:AB_2261274 | Immunofluorescence (IF; 1:20) |
| Antibody | Rat monoclonal anti-F4/80 | Biorad | Cat #MCA497 RRID:AB_2335599 | IF (1:100) |
| Antibody | Sheep polyclonal anti-digoxigenin-AP, Fab fragments | Roche | Cat #11093274910 RRID:AB_2313640 | In situ hybridization (ISH; 1:5000) Whole mount ISH (1:3000) |
| Recombinant DNA reagent | p-GEMt-Ctsk | This paper | | ISH probe |
| Recombinant DNA reagent | p-GEMt-Kazald1 | This paper | | ISH probe |
| Recombinant DNA reagent | p-GEMt-Krt17 | This paper | | ISH probe |
| Recombinant DNA reagent | Ctsk:mRuby3 | Backbone from *Geurtzen et al., 2022* | | Modified for this paper |
| Recombinant DNA reagent | Ctsk:eGFP | Backbone from *Geurtzen et al., 2022* | | Modified for this paper |
| Sequence-based reagent | *Ctsk* Fw primer | *Bryant et al., 2017* | PCR primer/in situ hybridization probe | GTGCAGAACCGACCCGATG |
| Sequence-based reagent | *Ctsk* Rv primer | *Bryant et al., 2017* | PCR primer/in situ hybridization probe | CAGCTGGACTCGGAGTGATGC |
| Sequence-based reagent | *Kazald1* Fw primer | *Bryant et al., 2017* | PCR primer/in situ hybridization probe | CTCGTGACATCCTGAGCCTGGAAG |
| Sequence-based reagent | *Kazald1* Rv primer | *Bryant et al., 2017* | PCR primer/in situ hybridization probe | GAAAATGGATAAGGTGGTGGGGAGGG |
| Sequence-based reagent | *Krt17* Fw primer | *Leigh et al., 2018* | PCR primer/in situ hybridization probe | CCTCTTGGACGTGAAGACC |
| Sequence-based reagent | *Krt17* Rv primer | *Leigh et al., 2018* | PCR primer/in situ hybridization probe | CCAGAGAAGATGAGCATACATCGG |
| Sequence-based reagent | *Ctsk* Fw primer | This paper | RT-qPCR primer | TGGCCCTTTTAACAACACCG |
| Sequence-based reagent | *Ctsk* Rv primer | This paper | RT-qPCR primer | ACTGAGTTGCAACAGCTTCC |
| Sequence-based reagent | *Rpl4* Fw primer | This paper | RT-qPCR primer | TGAAGAACTTGAGGGTCATGG |
| Sequence-based reagent | *Rpl4* Rv primer | This paper | RT-qPCR primer | CTTGGCGTCTGCAGATTTTTT |
| Sequence-based reagent | *Trap* Fw primer | This paper | RT-qPCR primer | TCATTGCCTGGTCAAGCATC |

*Continued on next page*

*Continued*

| Reagent type (species) or resource | Designation | Source or reference | Identifiers | Additional information |
|---|---|---|---|---|
| Sequence-based reagent | *Trap* Rv primer | This paper | RT-qPCR primer | TGGGCATAGTAGAACCGCAA |
| Sequence-based reagent | *Dcstamp* Fw primer | This paper | RT-qPCR primer | TGGAAACCAAAAGTGCAGCG |
| Sequence-based reagent | *Dcstamp* Rv primer | This paper | RT-qPCR primer | CCCCTCAGTGCCATCATTGT |
| Chemical compound, drug | Calcein | Sigma-Aldrich | Cat #C0875 | 0.1% solution |
| Chemical compound, drug | Alizarin red | Sigma-Aldrich | Cat #A5533 | 0.1% solution |
| Chemical compound, drug | Zoledronic acid | Sigma-Aldrich | Cat #SML0223 | Intraperitoneal (IP) injections, 200µg/kg |
| Commercial assay or kit | Click-iT EdU Cell Proliferation Kit for Imaging, Alexa Fluor 488 dye | Invitrogen | Cat #C10337 | IP injections, 10µg/g |
| Commercial assay or kit | Masson's trichrome staining | Sigma-Aldrich | Cat #HT15 | |
| Commercial assay or kit | pGEM-T Easy Vector Systems | Promega | Cat #A1360 | |
| Commercial assay or kit | RNAeasy Mini Kit | QIAGEN | Cat #74104 | |
| Software, algorithm | µManager | *Edelstein et al., 2014* | https://micro-manager.org/ | |
| Software, algorithm | Fiji | *Schindelin et al., 2012*, *Rueden, 2022* | https://github.com/fiji/fiji RRID:SCR_002285 | |
| Software, algorithm | Prism9 | GraphPad Software | https://www.graphpad.com | |
| Software, algorithm | Affinity Designer | Serif Europe | https://affinity.serif.com/ | |

## Axolotl husbandry and transgenesis

Axolotls (*A. mexicanum*) were maintained at the CRTD axolotl facility and at Harvard University. All procedures were performed according to the Animal Ethics Committee of the State of Saxony, Germany, and the Institutional Animal Care and Use Committee (IACUC) Guidelines at Harvard University (Protocol 11–32). Animals used were selected by its size (ST). Most experiments were done using animals 4–6 cm ST, unless stated otherwise. We performed experiments using white axolotls (*d/d*). In addition, we utilized transgenic lines shown in key resource table.

To generate the *Ctsk:mRuby3* or *Ctsk:eGFP* transgenic lines, a plasmid containing 4 kb of *Ctsk* promoter from zebrafish together with *Tol2* sequences was used (kind gift from Knopf Lab at CRTD) (*Geurtzen et al., 2022*). The *mRuby3* or *eGFP* coding region was cloned 3' from the promoter. For ligation, plasmid restriction was performed using the FseI and XhoI restriction enzymes (#R0588S, #R0146S, respectively; New England BioLabs, Frankfurt am Main, Germany). Fertilized embryos from *d/d* axolotls were injected with the *Ctsk:mRuby3* or *Ctsk:eGFP* vector and *Tol2* mRNA as previously described (*Khattak et al., 2014*). F0 animals were selected and grown in our colony until sexual maturity. For experiments, F0 were crossed with a *d/d* axolotl, and F1 animals were used.

## Experimental procedures in axolotls

For each experimental group, animals were randomly selected and assigned. Sample size was determined based on previous axolotl studies. The value of the biological replicates per experiment (n) is stated in each figure. For analyses of results, no outlier was removed.

In vivo skeletal staining was performed using calcein or alizarin red. A 0.1% solution was prepared for either dye with swimming water. Axolotls were submerged in solution for 5–10 min in the dark. After staining, animals were transferred to a tank with clean swimming water, which was changed as

many times until water was not stained. Amputations were performed either 10 min after staining or the next day for better visualization.

For amputations, animals were anesthetized with 0.01% benzocaine solution. All amputations were performed at the distal end of the calcified diaphysis using an Olympus SZX16 stereomicroscope. After surgical procedure, animals were covered with a wet tissue (with benzocaine) and allowed to recover for 10 min prior to be transferred back to swimming water. The FSF surgery was performed as described in *Tsai, 2020*; *Tsai et al., 2020*.

Zol treatment and EdU labeling were done by intraperitoneal injections in anesthetized axolotls. 200µg/kg of zol were injected every 3days (stock 1mg/mL in axolotl phosphate-buffered saline, APBS [80% PBS]). 10µg/g of EdU were injected 4hr prior to tissue collection (stock 2.5mg/mL in dimethyl sulfoxide, DMSO). Injection volume was adjusted to 10µL with APBS. After injections, animals were kept covered with a wet paper for 10min before returning them into the water tank.

In vivo imaging was performed in anesthetized animals. For stereoscope imaging, animals were placed in a 100 mm petri dish, and limb was positioned accordingly. An Olympus SZX16 stereoscope microscope (objective: SDF Plapo 1xPF) was used. For confocal imaging, animals were place in a glass bottom dish (ø: 50/40 mm, #HBSB-5040, Willco Wells, Amsterdam, The Netherlands). A wet tissue with benzocaine was laid on top of the animal to avoid it to dry, and a silica block was laid on top of the hand to flatten it and improve light penetrance. A Zeiss confocal laser scanning microscope LSM780 (objectives: Plan apochromat 10×/0.45 or Plan-apochromat 20×/0.8) was used.

For tissue collection, animals were anesthetized prior to collection. After it, animals were euthanized by exposing them to lethal anesthesia (0.1% benzocaine) for at least 20 min. Tissue fixation and further procedures are described specifically for each case.

## Paraffin sectioning

Limbs were isolated and fixed with MEMFa 1× (MOPS 0.1M pH 7.4/EGTA 2mM/ MgSO$_4$ × 7 H$_2$O 1mM/3.7% formaldehyde) overnight at 4°C. Samples were washed with PBS and dehydrated with serial EtOH washes (25, 50, 70, and ×3 100%). Samples were then incubated three times with Roti-Histol (#6640, Carl Roth, Karlsruhe, Germany) at RT and four times with paraffin (Roti-Plast, #6642, Carl Roth) at 65°C in glass containers. After last incubation, samples were embedded in paraffin using plastic containers and stored at RT. Longitudinal sections of 6µM thickness were obtained.

## Cryosectioning

Limbs fixed with MEMFa were washed with PBS and decalcified with EDTA 0.5 M at 4°C for 48 hr. Next, limbs were washed with PBS and incubated overnight with sucrose 30% at 4°C. Samples were embedded in O.C.T. compound (#4583, Tissue-Tek, Umkirch, Germany) using plastic molds and frozen with dry ice for 1 hr prior to storage at –20°C. Longitudinal sections of 12 µm thickness were cut and mounted on superfrost slides. Slides were kept at –20°C until processed.

## TRAP enzymatic staining

TRAP enzymatic staining was performed in cryosections. Slides were dried for 1 hr prior to wash them with PBS + 0.1% Tween-20 for 10 min. Next, slides were permeabilized with PBS + 0.3% Tx-100 for 1 hr. After permeabilization, slides were equilibrated by three washes with TRAP buffer (NaAcetate 0.1 M/acetic acid 0.1 M/NaTartrate 50 mM/pH 5.2) for 10 min at 37°C in water bath. Slides were stained with color reaction buffer (TRAP buffer/Naphthol AS-MX phosphate 1.5 mM/Fast Red Violet LB Salt 0.5 mM) for 1 hr at 37°C in water bath. After staining, slides were washed three times with PBS for 10 min and mounted with Entellan (#1.07960, Sigma-Aldrich). Images were taken in a Zeiss Axio Observer.Z1 inverted microscope.

## Immunofluorescence

For IF, cryosections were used. Slides were dried at RT for at least 1 hr. Sections were washed three times with PBS + 0.3% Tx-100 prior to blocking with PBS + 0.3% Tx-100 + 10% normal horse serum (NHS, #S-2000–20, Vector Labs, Burlingame, CA, USA) for 1 hr. Primary anti-CTSK or anti-F4/80 antibody incubation was done in blocking solution for 1 hr at RT and then overnight at 4°C. Sections were then washed three times with PBS + 0.3% Tx-100 and incubated with Goat anti-Rabbit, Alexa Fluor 647 antibody (1:200, #A-21245, Invitrogen, RRID:AB_2535813) and Hoechst 33342 1:1000 for

2 hr. Finally, sections were washed three times with PBS + 0.3% Tx-100 and mounted using Mowiol mounting medium (#0713 Carl Roth). Imaging was performed on a Zeiss Axio Observer.Z1 inverted microscope with an ApoTome1 system (objectives: Plan-apochromat 10×/0.45 or Plan-apochromat 20×/0.8).

## Masson's trichrome staining

Masson's trichrome staining on paraffin sections or cryosections was performed following the producer's recommendations (Procedure No. HT15, Sigma-Aldrich). Imaging was performed in a Zeiss Axio Observer.Z1 inverted microscope (objective: Plan-apochromat 20×/0.8). For performing the staining in *Ctsk:eGFP* sections (*Figure 5—figure supplement 2*), we first mounted the slides in glycerol/PBS 1:1 + Hoechst 33342 1:10,000 and imaged them. Coverslip was removed by incubating slides in PBS for 1hr and then stained with Masson's trichrome as previously stated.

## RNA probes for in situ hybridization

*Ctsk, Kazald1,* and *Krt17* probes were created by TA cloning. Probe amplification was done using primers previously published. Ligation was done into a pGEM-T easy vector system I. To confirm successful cloning, vectors were purified and sequenced using the Mix2Seq Kit (Eurofins Genomics, Ebersberg, Germany).

For synthesizing the ISH probes, in vitro transcription was carried out using a T7 polymerase (#RPOLT7-RO, Roche, Mannheim, Germany) or a SP6 polymerase (#RPOLSP6-RO, Roche), following provider's instructions. Prior to transcription, 5 µg of plasmid were linearized using the restriction enzyme SpeI-HF (#R3133S, New England BioLabs) for *Ctsk* and *Krt17*, or SphI-HF (#R3182S, New England BioLabs) for *Kazald1*. Probes were purified using the RNAeasy Mini Kit according to provider's instructions.

## In situ hybridization

ISH was performed in cryosections using *Ctsk or Kazald1* probe following a previously published protocol (*Knapp et al., 2013*). When the protocol was finished, slides were fixed in formaldehyde 4% overnight at RT. Slides were then dehydrated with serial EtOH washes (25, 50, 70, and 100%) prior to wash with RotiHistol and mounting with Entellan. Imaging was performed on a Zeiss Axio Observer. Z1 inverted microscope.

## Whole mount in situ hybridization

WISH was performed using *Krt17* probes. Protocol was adapted from *Woltering et al., 2009*. Briefly, samples were dehydrated with serial MetOH washes (25, 50, and 75% in PBS + 0.1% Tween-20 and 100%). Limbs were bleached in MetOH + 6% $H_2O_2$ at RT and then rehydrated with serial washes of MetOH. Then, limbs were washed with TBST (1× TBS, 0.1% Tween-20) and treated with 20µg/mL proteinase K in TBST for 30min at 37°C. After incubation, limbs were washed with TBST and rinsed with trietanolamine 0.1M pH 7.5 (#90278, Sigma-Aldrich). Limbs were incubated with freshly prepared 0.1M TEA+ 1% acetic anhydride (#320102, Sigma-Aldrich) for 10min and then washed again with TBST. Next, limbs were fixed with 4% paraformaldehyde (PFA)+ 0.2% glutaraldehyde (#G6257, Sigma-Aldrich) for 20min and washed with TBST. TBST was removed, and limbs were incubated with previously warmed Pre-Hyb solution (hybridization buffer without probe) at 60°C for 4hr, prior to be transferred into pre-warmed hybridization buffer+ probe (6µL/mL) and incubated overnight at 60°C. The next day, limbs were washed at 60°C with pre-warmed 5× saline-sodium citrate(SSC) solution twice for 30min, with 2× SSC solution three times for 20min, and with 0.2× SSC three times for 20min. Limbs were then washed with Tris-sodium-EDTA (TNE) solution twice for 10min at RT prior to incubation with 20µg/mL RNAse A in TNE solution for 15min. After incubation, limbs were washed with TNE solution twice for 10min, and with maleic acid buffer (MAB) solution three times for 5min. Limbs were blocked with MAB solution+ 1% blocking reagent for 1.5hr and then incubated with MAB solution+ 1% blocking reagent+ 1:3000 anti-digoxigenin-AP, Fab fragments for 4hr at RT. Next, limbs were washed with MAB solution three times for 5min each and then overnight at RT. On day 3, limbs were washed with MAB solution five times for 1hr each and again overnight. After MAB washes, limbs were washed with NTMT three times for 10min at RT and then incubated with freshly made alkaline phosphatase buffer (NTMT)+ 20µL/mL NBT/BCIP (#11681451001, Roche) for 4–6hr. Reaction was then

stopped by incubating with PBS + 0.1% Tween-20 twice for 10min and then fixing with 4% PFA at 4°C overnight. After fixation, limbs were washed with PBS + 0.1% Tween-20 and stored in that solution at RT. Imaging was performed on a Zeiss Discovery.V20 stereomicroscope.

## Alcian blue/alizarin red staining

Staining was performed as recently described (*Riquelme-Guzmán et al., 2021*). Imaging was performed on a Zeiss Discovery.V20 stereomicroscope (objective: Plan S 1.0×).

## EdU staining (whole mount and in cryosections)

Limbs from axolotls injected with EdU were fixed with MEMFa 1× overnight at 4°C and then washed with PBS. For whole mount EdU staining, limbs were washed with PBS + 0.3%Tx-100 twice for 2hr and then blocked with PBS + 0.3%Tx-100 + 5%goat serum+ 10%DMSO for 24hr at RT. Click-iT EdU cell proliferation kit, Alexa Fluor 488 was used following provider's instructions. Samples were incubated in reaction cocktail for 4hr at RT. After incubation, samples were washed with PBS + 0.3%Tx-100 four times for 15min. Next, samples were incubated with TO-PRO-3 1:10,000 in PBS + 0.3%Tx-100 for 1hr at RT. Finally, limbs were washed with PBS four times for 15min each at RT. Limbs were cleared by dehydration with serial washes of EtOH (25, 50, 70, 100%) for 2hr each at 4°C. Samples were then incubated overnight in 100% EtOH at 4°C prior to clearing with ethyl cinnamate (#112372, Sigma-Aldrich) at RT for at least 2hr. Samples were imaged the same day on a Zeiss confocal laser scanning microscope LSM 780 (objectives: Plan-apochromat 20×/0.8).

For cryosections, provider's instructions were followed, but Hoechst 33342 1:1000 was used as DNA dye. Samples were then washed three times with PBS + 0.3% Tx-100 and mounted using Mowiol mounting medium. Imaging was performed on a Zeiss Axio Observer.Z1 inverted microscope (objectives: Plan-apochromat 20×/0.8).

## RNA purification and RT-qPCR

Limbs for RNA isolation were stored in RNA*later* (#AM7024, Invitrogen) at –20°C until all samples were collected. RNA isolation was performed using the RNAeasy Mini Kit. 50ng of RNA were used for cDNA synthesis using the PrimeScript RT reagent Kit (#RR037A, Takara, Göteborg, Sweden) following the provider's instructions. RT-qPCR was performed using the TB Green Premix Ex Taq (Tli RNAseH Plus) kit (#RR420A, Takara). RT-qPCR was done using a LightCycler 480 system with a pre-defined protocol for SYBR Green. Results were analyzed using the ΔΔCT method and the *Rpl4* housekeeping gene. After analysis, results were shown as relative levels compared to a control.

## Polarization microscopy

The LC-PolScope is a powerful tool to quantitatively image optically anisotropic materials having a refractive index that depends on the polarization and propagation of light (birefringence), such as collagen, the main component of cartilage ECM (*Fox et al., 2009*). An LC-PolScope (on a Ti Eclipse microscope body) with a sCMOS camera (Hamamatsu Orca Flash 4.0) was used. Acquisitions were done with a 20×/0.8 objective and using µManager software (*Edelstein et al., 2014*). Two images were acquired: the retardance and the slow axis orientation. The retardance correlates with the amount of birefringent components, while the slow axis orientation image provides information on the orientation of those components, i.e. the angle in which they are aligned in the sample. Retardance and slow axis orientation images were aligned using a custom-made MATLAB script such that the x-axis corresponded to the proximodistal axis and the y-axis corresponded to the anteroposterior axis, with the elbow on the top-right corner of the image. The angle was measured with respect to the proximodistal axis. Once the images were aligned, the regions of interest were cropped and segmented using the Trainable Weka Segmentation plugin from Fiji (*Arganda-Carreras et al., 2017*). The segmentation was done to obtain masks for the collagen regions and to remove the cells from the analysis. The masks were then applied to the slow axis orientation images, and the orientations of the collagen fibers were quantified using MATLAB.

## Curating RNA-seq data

Recently published axolotl RNA-seq datasets (*Tsai et al., 2020*; *Tsai et al., 2019*) were used to evaluate osteoclast-related transcript levels in samples upon FSF surgery. For curating the datasets, R

**Table 1.** Criteria for scoring integration in limbs at 45 dpa.

| Question | Yes (value) | No (value) |
| --- | --- | --- |
| Is there angulation present? | +1 | +0 |
| Is there superposition between the mature tissue and the regenerated structure? | +1 | +0 |
| Does regeneration result in element doubling? | +1 | +0 |
| Does the regenerated structure constitute one continuous element? | +0 | +1 |

Studio was used (RStudio Team, http://www.rstudio.com/). In order to find osteoclast-related transcript identifiers, the human, mouse, and *Xenopus* protein orthologous for each transcript were used. With the protein sequences, a protein blast was performed using the predicted proteins from Bryant et al. de novo axolotl transcriptome (supplementary data *Bryant et al., 2017*). The best three matches for each protein were used for browsing in Tsai's transcriptome. The datasets from both of Tsai et al. works were combined and filtered in order to have only the results from 0 dpa, 5 dpa, and 5 dpa in FSF surgery. In addition, the 2N, 4N, and EP fractions at 5 dpa were also filtered from the combined datasets. For organizing, filtering, and calculating z-scores in the datasets, the tidyverse package (*Wickham et al., 2019*) and plyr package (*Wickham, 2011*) were used. For creating the heatmaps, the ggplot2 package was used (*Wickham, 2016*).

To find possible candidates involved in osteoclast recruitment and differentiation, a search in the available literature was done for each differentially downregulated transcript in FSF samples (385 transcripts, *Tsai et al., 2020* supplementary data). Transcripts, which have been connected to osteoclast function or belong to a protein family shown to play a role in osteoclast recruitment and differentiation, were filtered and heatmaps were created for better visualization of the levels during regeneration and in the different fractions (*Figure 6C and D*).

### Score matrix for integration phenotypes

To assess the integration phenotype in limbs at 45 dpa (*Figure 4*, *Figure 4—figure supplement 1*, *Figure 4—source data 1*), we assessed each limb individually using the criteria shown in *Table 1*. The maximum final score for a malformed skeletal element was 4. Conversely, in the best regenerated structures, the final score was 0.

### μCT scan

Scans were performed as recently described (*Riquelme-Guzmán et al., 2021*). Threshold was set to 220 mg HA/cm$^3$.

### Statistical analysis

Statistical analyses were performed using the software Prism9 (GraphPad Software, LLC, San Diego, CA, USA) for macOS. Statistical tests performed are described in each figure. p-Values <0.05 were considered statistically significant.

### Image processing and figure design

All images were processed using Fiji (*Schindelin et al., 2012*). Processing involved selecting regions of interest, merging or splitting channels, and improving brightness levels for proper presentation in figures. Maximum intensity projections were done in some confocal images, and it is stated in the respective figure's descriptions. Stitching of tiles was done directly in the acquisition software Zen (Zeiss Microscopy, Jena, Germany). Figures were created using Affinity Designer (Serif Europe, West Bridgford, UK).

## Acknowledgements

We thank past and current members of the Sandoval-Guzmán lab for continuous support during the development of this work. We are also grateful to Anja Wagner, Beate Gruhl, and Judith Konantz for

their impeccable dedication to the axolotl care. We also thank Karina Geurtzen and Franziska Knopf for providing the plasmid containing the *Ctsk* promoter from zebrafish. We thank Can Aztekin for helpful discussions and comments on the manuscript. This work was funded by a DFG Research Grant (432439166). CRG was supported by the DIGS-BB Fellow award. This work was supported by the Light Microscopy Facility, a Core Facility of the CMCB Technology Platform at TU Dresden.

## Additional information

### Funding

| Funder | Grant reference number | Author |
|---|---|---|
| Deutsche Forschungsgemeinschaft | SA 3349/3-1 | Camilo Riquelme-Guzmán |
| DFG Research Grant | 432439166 | Tatiana Sandoval-Guzmán |

The funders had no role in study design, data collection and interpretation, or the decision to submit the work for publication.

### Author contributions

Camilo Riquelme-Guzmán, Conceptualization, Data curation, Formal analysis, Investigation, Visualization, Methodology, Writing – original draft, Writing – review and editing; Stephanie L Tsai, Data curation, Investigation, Methodology, Writing – review and editing; Karen Carreon Paz, Congtin Nguyen, Data curation, Investigation, Visualization; David Oriola, Data curation, Software, Formal analysis, Investigation, Visualization, Methodology, Writing – review and editing; Maritta Schuez, Investigation, Methodology; Jan Brugués, Resources, Methodology, Writing – review and editing; Joshua D Currie, Conceptualization, Supervision, Investigation, Methodology, Writing – review and editing; Tatiana Sandoval-Guzmán, Conceptualization, Formal analysis, Supervision, Funding acquisition, Writing – original draft, Project administration, Writing – review and editing

### Author ORCIDs

Camilo Riquelme-Guzmán (iD) http://orcid.org/0000-0002-5126-6584
Stephanie L Tsai (iD) http://orcid.org/0000-0001-7549-3418
David Oriola (iD) http://orcid.org/0000-0002-8356-7832
Jan Brugués (iD) http://orcid.org/0000-0002-6731-4130
Tatiana Sandoval-Guzmán (iD) http://orcid.org/0000-0003-1802-5145

### Ethics

All procedures were performed according to the Animal Ethics Committee of the State of Saxony, Germany, and the Institutional Animal Care and Use Committee (IACUC) Guidelines at Harvard University (Protocol 11-32).

### Decision letter and Author response

Decision letter https://doi.org/10.7554/eLife.79966.sa1
Author response https://doi.org/10.7554/eLife.79966.sa2

## Additional files

### Supplementary files
• MDAR checklist

### Data availability
No dataset have been generated for this manuscript.

The following previously published datasets were used:

| Author(s) | Year | Dataset title | Dataset URL | Database and Identifier |
| --- | --- | --- | --- | --- |
| Tsai SL, Baselga-Garriga C, Melton DA | 2020 | Wound epidermis-dependent transcriptional programs | https://www.ncbi.nlm.nih.gov/geo/query/acc.cgi?acc=GSE132317 | NCBI Gene Expression Omnibus, GSE132317 |
| Tsai SL, Baselga-Garriga C, Melton DA | 2019 | Blastemal progenitors modulate immune signaling during early limb regeneration | https://www.ncbi.nlm.nih.gov/geo/query/acc.cgi?acc=GSE111213 | NCBI Gene Expression Omnibus, GSE111213 |

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
