## [Editor Report]

Using a well-established and elegant axolotl limb regeneration model and transgenic reporter strains, the study reveals the potential role of osteoclast-mediated resorption in limb regeneration. The findings from the work provide a new understanding of how the skeleton is primed for regeneration.

---

## [Decision Letter]

**Decision letter after peer review:**

Thank you for submitting your article "Osteoclast-mediated resorption primes the skeleton for successful integration during axolotl limb regeneration" for consideration by *eLife*. Your article has been reviewed by 3 peer reviewers, and the evaluation has been overseen by a Reviewing Editor and Kathryn Cheah as the Senior Editor. The following individual involved in the review of your submission has agreed to reveal their identity: Andrei S Chagin (Reviewer #1).

Essential revisions:

Overall, this is a well-executed and technically sound study. However, with the current set of data, the correlation between bone resorption and bone regeneration is not fully established, and additional experiments/analyses are needed. Particularly, additional analyses of zol-treated or full skin flap surgery-treated groups would strengthen this study. In addition, the authors should consider addressing several other concerns raised by the reviewers.

*Reviewer #1 (Recommendations for the authors):*

Dear authors, the manuscript is nicely done, and the data are well presented. In my opinion, the intrinsic limitation of the study is the question, on which this work is focused on. It is of limited value, in my opinion. Nevertheless, here are some specific comments and suggestions to improve the study.

There are some questions, which can make the story more exciting.

1. Bisphosphonates block bone resorption but osteoclasts remain there. What would be the differences in the regenerative process in the case of osteoclast ablation versus inhibition of their resorption ability? Is the same outcome anticipated? What is the role of osteoclasts in the regeneration of urodeles?

2. Tissue histolysis causes degradation of the extracellular matrix. What does happen with bone cells upon bone resorption? Do they dedifferentiate upon matrix destruction, do they die, and do they contribute to new bone formation?

3. Reintegration of old and new bone is a very interesting process. How does it occur on a cellular level? Is there intramembranous or endochondral ossification, where and what matrix starts to be built, and by what cells?

There are a few uncertainties in the text, which would be good to clarify.

1. Lines 226-228. The statistic and p values supporting this statement would be good to present: The regenerated skeleton formed at an adjacent plane, therefore lacking continuity with the pre-existing skeleton.

2. Lines 230-234: Does this statement refer to zol-treated or untreated animals? Please clarify.

3. Lines 239-240. Please clarify if the statement refers to zol-treated or untreated animals. Please clarify how the original axis was determined.

4. Lines 251-253. Please clarify if the statement refers to zol-treated or untreated animals.

5. Line 283. Please clarify WISH abbreviation, the first time mentioned in the text to avoid the reader searching method section.

6. Please clarify if Figure 5e,f,h depict limbs upon flap-surgery or normal regenerating limb.

7. Line 320 says (red arrows, Figure 6C). I did not find red arrows anywhere in Figure 6.

8. The paragraph presented between lines 314-326 is rather associative (see my major comments above). Please present accordingly.

9. Lines 352-360. Double staining is needed to make a reliable conclusion (see my major comments above). Please soften conclusions if not performed.

*Reviewer #2 (Recommendations for the authors):*

Overall this is a very nice manuscript with new and strong data. The work is well executed and it is a pleasure to read.

Reviewer #3 (Recommendations for the authors):

1. Abstract:

– I feel that the abstract is oversimplified and overstated without important details. The authors should at least clarify that they used zol to inhibit bone resorption. Also, I do not see any data showing that the extent of resorption is directly correlated with the extent of integration in a quantitative manner. It is also not clear how these data demonstrate that the wound epithelium creates "a zone of influence" and functions as a major regulator of skeletal resorption. It seems to be a mere correlation.

2. Introduction:

– The introduction is overall well-written but is also too long. The authors perhaps try to cover too much in this section. They could make this section more succinct.

3. Results:

– Line 161: CTSK may also be expressed by mesenchymal cells in the periosteum. The authors should acknowledge the possibility that these cells may include cells other than osteoclasts.

– Line 171: Can the authors clarify if they also used in vivo imaging for these elegant transgenic assays?

– I feel that the logical connection between the section prior to Line 228 and the section after Line 229 is weak. It would be important to show how the angulation is also altered in zol-treated animals. It is possible that this phenotype is completely independent of zol-treatment.

– Line 264: Again, it is possible that these CTSK^+^ cells include periosteal cells, which may have an important contribution to blastema formation.

– For Figure 5E-H, it would be important to show how the distribution of osteoclasts changes when FSF is performed.

– Figure 6 is interesting but very descriptive. The authors should at least provide some validation of these predicted changes.

– Figure 7: What is the significance of decreased blastema formation in zol-treated animals? The limb regeneration occurs anyway, only associated with a relatively minor defect in skeletal integration. The authors would need to perform EdU assays in the blastema of zol-treated animals.

– Line 366: aggrupation (?)

– In Figure 7D, the *Sox9* reporter assay is nice, but it would be important to see how these cells are disrupted in zol-treated animals, to enhance the relevance of this experiment.

4. Discussion:

– Line 411: I do not see any convincing quantitative data that support this statement, "the more resorption, the better the integration. Evidence now is somewhat anecdotal.

[Editors’ note: further revisions were suggested prior to acceptance, as described below.]

Thank you for resubmitting your work entitled "Osteoclast-mediated resorption primes the skeleton for successful integration during axolotl limb regeneration" for further consideration by *eLife*. Your revised article has been evaluated by Kathryn Cheah (Senior Editor) and a Reviewing Editor.

The manuscript has been improved but there are some remaining issues that need to be addressed, as outlined below:

1) The authors should clarify in the abstract that they used zoledronic acid to inhibit bone resorption.

2) It is necessary for the authors to include a new paragraph in the discussion to talk about the potential pitfalls of the study, especially the possible effect of zol on macrophages and that Ctsk-gfp may also visualize blastema cells.

*Reviewer #1 (Recommendations for the authors):*

The manuscript has improved upon revision. However, my major concern stays the same if not get worsened. Specifically – do osteoclasts have any role in the presented story. Please see my arguments below.

The author did some additional analysis in relation to this issue and said the following in their rebuttal letter "However, to our surprise, it only resulted in a decreased number and function inhibition. ". However, the data were not presented in the revised manuscript.

It further rises my suspicion that osteoclasts may not be a key player involved in the observed reintegration defects.

Here are additional observations and arguments making me so concerned:

Macrophages are unequivocal for limb regeneration in salamanders since their ablation abrogates limb regeneration entirely (PNAS, 2013, 110 (23) 9415-9420). Zoledronic acid may directly affect macrophages (numerous references, i.e., Cellular Oncology volume 36, pages 505-514 (2013)) including polarization toward M1 (Faseb J 2019 Apr;33(4):5208-5219).

Thus, the observed effect of impaired reintegration, may not be related to osteoclasts activity but to macrophages.

Furthermore, the developmental stage of axolotl used in this study is 4-6 cm snout-tail (method section) at the time of limb amputation. At this stage neither ulna nor radius are yet developed bone tissue, only cartilage. Accordingly, the observed labeling with calcein reflects calcification of hypertrophic area prior to ossification starts.

Thus, the observed changes in the calcified area may not necessarily reflect osteoclast activity or skeletal resorption but calcium loss due to numerous other reasons such as acidic pH of the inflammatory environment, active remodeling of extracellular matrix by MMP8/13 proteases (ref Gerber et al., Science 2018), etc.

In this same paper, a single-cell RNAseq of axolotl limbs and blastema has been performed. Analysis of this dataset reveals that Ctsk is highly expressed by a large fraction of blastema cells, particularly those of mesenchymal origin, i.e., Prrx1-sorted.

Thus, the observed Ctsk-GFP positive cells may not be osteoclasts but blastema-forming cells, whereas white arrowheads on histological images (Figure 5F,G) may well point toward macrophages.

Thus, altogether, the observed effects of tissue reintegration may have no relation to osteoclasts at all. It puts the entire story under question.

*Reviewer #2 (Recommendations for the authors):*

The authors have addressed all of my concerns raised in my previous review.

In my opinion this manuscript is ready for publication.

*Reviewer #3 (Recommendations for the authors):*

The authors revised the manuscript in response to the comments from three reviewers during the previous round of peer review. The authors performed additional experiments to support the conclusion further. Below, I am going to evaluate the authors' responses to my comments.

1. The authors did not clarify that they used zoledronic acid to inhibit bone resorption in the abstract, as requested. It is extremely important to specify in the abstract that they used zol, but not other reagents, to inhibit resorption. This is an essential piece of information to support the overall integrity of the work.

2. The authors trimmed down the introduction for approximately 9 lines.

3. The authors convincingly showed that CTSK:eGFP is also expressed in peri-skeletal cells.

4. The authors now included a quantitative assessment of the integration outcomes in regenerated limbs, showing the relationship between the integration score and the resorption rate. This data is highly supportive of the conclusion.

5. The authors did not evaluate the *Sox9* reporter in zol-treated mice, as requested.

6. The authors quantified the number of EdU+ cells in the blastema of zol-treated animals, as requested.

7. The authors did not provide validation for re-analysis of a published RNA-seq dataset. The authors could have used in situ hybridization to validate the predicted gene expression changes.

Overall, I feel that the authors managed to improve the manuscript within a given time frame to support their conclusion.

---

## [Author Response]

Reviewer #1 (Recommendations for the authors):Dear authors, the manuscript is nicely done, and the data are well presented. In my opinion, the intrinsic limitation of the study is the question, on which this work is focused on. It is of limited value, in my opinion. Nevertheless, here are some specific comments and suggestions to improve the study.There are some questions, which can make the story more exciting.

We sincerely thank Reviewer #1 for their comments and suggestions, we do believe the suggestions have made the work more exciting. We hope that with the initially submitted and new data we are able to show that although it is tempting to extrapolate our knowledge from other systems, species-specific mechanisms are a reality. As we mention in this manuscript, the fate of osteoclast is not completely understood, and in the case of regeneration, the very fast arrival and clearance is in itself fascinating and worth exploring.

1. Bisphosphonates block bone resorption but osteoclasts remain there. What would be the differences in the regenerative process in the case of osteoclast ablation versus inhibition of their resorption ability? Is the same outcome anticipated?

Taking into consideration works in the literature, we expected that osteoclasts would be ablated upon zoledronic acid treatment. However, to our surprise, it only resulted in a decreased number and function inhibition. The variability in tissue resorbed in normal conditions reported in our work gives us already a range of integration phenotypes, which are exacerbated when we applied zoledronic acid. Upon osteoclasts ablation, we speculate that a more consistent lack of resorption would be observed and thus a higher number of dramatic phenotypes.

It is important to note that osteoclasts have been reported to influence other processes, such as vascularization, and in the context of osteoclasts ablation, those processes could be examined in detail using diverse tools such as new transgenic lines. Finally, during regeneration, resorption of the amputated skeleton occurs in a short time window where the regenerative process is occurring simultaneously. Thus, we consider that complete ablation or decreased numbers, will unambiguously show us an effect on tissue integration. We believe in this report we provide the first in-depth assessment of a histolytic process and its consequences for limb regeneration in the axolotl.

What is the role of osteoclasts in the regeneration of urodeles?

We hypothesize that the role of osteoclasts in regeneration is to clear the damaged skeleton and to prime it to receive the newly formed skeleton and amalgamate the old matrix to the new one. Other roles could include the freeing of cells to activate secretory activity, however, to answer this question we will require further studies.

2. Tissue histolysis causes degradation of the extracellular matrix. What does happen with bone cells upon bone resorption? Do they dedifferentiate upon matrix destruction, do they die, and do they contribute to new bone formation?

Several works showed the lack of participation of skeletal cells in salamander regeneration (Currie et al., 2016; McCusker et al., 2016). In addition, a previous work has shown that chondrocytes at the very amputation plane do divide however do not migrate into the blastema (Currie et al., 2016).

Due to the magnitude of the skeletal resorption that we describe here, we hypothesized that skeletal cells would either die through apoptosis or be phagocytosed by immune cells, likely Ctsk^+^ cells. In the past, we had performed stainings for Caspase 3, where there was minimal signal. However, to be more conclusive and to address the reviewer’s question, we performed TUNEL assay at 5 and 7 dpa and see very few positive cells in the skeleton, indicating DNA cleavage/damage (Figure 2 —figure supplement 1D). Additionally, we performed imaging of amputated digits from Ctsk^+^ /*Sox9*^+^ animals. We observed that *Sox9*^+^ chondrocytes at the site of resorption are phagocytosed by Ctsk^+^ cells (Figure 2E, F). The minimal apoptosis inside the skeleton is supported by a previous work in which apoptosis was mostly observed in the wound epithelium, muscle and periosteum (Bucan et al., 2018)

Currie JD, Kawaguchi A, Traspas RM, Schuez M, Chara O, Tanaka EM (2016) Live Imaging of Axolotl Digit Regeneration Reveals Spatiotemporal Choreography of Diverse Connective Tissue Progenitor Pools. Developmental Cell 39:411–423. https://doi.org/10.1016/j.devcel.2016.10.013

McCusker CD, Diaz-Castillo C, Sosnik J, Phan AQ, Gardiner DM (2016) Cartilage and bone cells do not participate in skeletal regeneration in Ambystoma mexicanum limbs. Developmental Biology 416:26–33. https://doi.org/10.1016/j.ydbio.2016.05.032

Bucan V, Peck C-T, Nasser I, Liebsch C, Vogt PM, Strauß S (2018) Identification of axolotl BH3-only proteins and expression in axolotl organs and apoptotic limb regeneration tissue. Biology Open 7:bio036293-9. https://doi.org/10.1242/bio.036293

3. Reintegration of old and new bone is a very interesting process. How does it occur on a cellular level? Is there intramembranous or endochondral ossification, where and what matrix starts to be built, and by what cells?

The core of the work was assessing regeneration in animals up to 6 cm total length. However, we confirmed that resorption occurs in adult animals, with ossified bone in the limbs. From our experience working with axolotls, in adult and metamorphic animals, regeneration occurs by first forming a chondrogenic anlage (endochondral ossification). In juvenile axolotls, the appendicular skeleton is composed solely of cartilage and thus its regeneration occurs by condensation of chondroprogenitors (as shown in Figure 7D). It is unclear what type of matrix starts to be built and the cells secreting it, although we hypothesize that condensation of chondroprogenitors might activate developmental pathways which initiate the deposition of cartilage ECM (collagen type II for example). The activation of developmental pathways during regeneration of connective tissue has already been demonstrated in a previous work (Gerber, et al. 2018). Also, we have recently shown that the viscoelasticity recovery of the cartilage is progressive during regeneration and surprisingly, at 30 dpa has not reach the original values, although morphologically, the tissue is comparable (Riquelme-Guzman et al. 2022). This underlines the need to investigate the exact cellular mechanism by which the cartilage and its matrix are restored.

Gerber T, Murawala P, Knapp D, Masselink W, Schuez M, Hermann S, Gac-Santel M, Nowoshilow S, Kageyama J, Khattak S, Currie J, Camp JG, Tanaka EM, Treutlein B (2018) Single-cell analysis uncovers convergence of cell identities during axolotl limb regeneration. Science 20:eaaq0681-19. https://doi.org/10.1126/science.aaq0681

Riquelme-Guzmán C, Beck T, Edwards-Jorquera S, Schlüßler R, Müller P, Guck J, Möllmert S, Sandoval-Guzmán T (2022) in vivo assessment of mechanical properties during axolotl development and regeneration using confocal Brillouin microscopy. Open Biol 12:220078. https://doi.org/10.1098/rsob.220078

There are a few uncertainties in the text, which would be good to clarify.1. Lines 226-228. The statistic and p values supporting this statement would be good to present: The regenerated skeleton formed at an adjacent plane, therefore lacking continuity with the pre-existing skeleton.

We thank the reviewer for their suggestion. To have a more accurate assessment of the integration phenotypes, we developed a score matrix that took into consideration various aspects of tissue integration (a clear description can be found in the Materials and methods). With this, we were able to quantify the integration phenotypes observed and correlate them with the resorption rate for each element individually. As shown in line 283-290 and Figure 4C, we observed a statistically significant correlation between resorption rate and integration success.

2. Lines 230-234: Does this statement refer to zol-treated or untreated animals? Please clarify.

This section was replaced for the new analysis of integration.

3. Lines 239-240. Please clarify if the statement refers to zol-treated or untreated animals. Please clarify how the original axis was determined.

Statement was clarified. “Untreated” was added in line 292. Moreover, the image axis was clarified in the figure description.

4. Lines 251-253. Please clarify if the statement refers to zol-treated or untreated animals.

Statement was clarified. “Untreated” was added in line 300

5. Line 283. Please clarify WISH abbreviation, the first time mentioned in the text to avoid the reader searching method section.

Abbreviation was modified for clarity.

6. Please clarify if Figure 5e,f,h depict limbs upon flap-surgery or normal regenerating limb.

“Untreated” was added in line 366 for clarity of experiments.

7. Line 320 says (red arrows, Figure 6C). I did not find red arrows anywhere in Figure 6.

“Red arrows” was deleted.

8. The paragraph presented between lines 314-326 is rather associative (see my major comments above). Please present accordingly.

We agreed with the Reviewer’s comment. The paragraph between now lines 397-420 was modified to present results more accurately.

9. Lines 352-360. Double staining is needed to make a reliable conclusion (see my major comments above). Please soften conclusions if not performed.

We thank the reviewer for their suggestion. To have a more reliable conclusion, we have replaced Figure 7C with a representative image of ISH for Kazald1 in a section from a limb at 7 dpa. There, the position of Kazald1^+^ cells next to the skeletal element is clearly visible, supporting our observations that blastema starts to form at the site where resorption occurs. Moreover, we have added Figure 7-supplement 1C, in which osteoclasts (identified by our transgenic line Ctsk:eGFP) are imaged at the same optical plane as chondrocytes which are forming the new phalanx (identified by our transgenic line *Sox9*:*Sox9*-T2a-mCherrynls). Overall, we believe our conclusions are sustained. We proved that blastema cells started to appear before resorption, under the amputation plane. Additionally, cartilage starts to regenerate before resorption finished and while osteoclasts are still present in the tissue.

Reviewer #2 (Recommendations for the authors):Overall this is a very nice manuscript with new and strong data. The work is well executed and it is a pleasure to read.

We sincerely thank Reviewer #2 for their comments and suggestions.

Reviewer #3 (Recommendations for the authors):1. Abstract:– I feel that the abstract is oversimplified and overstated without important details. The authors should at least clarify that they used zol to inhibit bone resorption. Also, I do not see any data showing that the extent of resorption is directly correlated with the extent of integration in a quantitative manner. It is also not clear how these data demonstrate that the wound epithelium creates "a zone of influence" and functions as a major regulator of skeletal resorption. It seems to be a mere correlation.

Taken together the suggestions from all reviewers, we have improved and strengthen the conclusions from this work. We have provided a more quantitative assessment of the integration phenotypes. Nevertheless, we have made some modifications to the abstract to better demonstrate the results from this work.

2. Introduction:– The introduction is overall well-written but is also too long. The authors perhaps try to cover too much in this section. They could make this section more succinct.

We can see how the introduction could be felt as lengthy and thus, we reduced the content.

3. Results:– Line 161: CTSK may also be expressed by mesenchymal cells in the periosteum. The authors should acknowledge the possibility that these cells may include cells other than osteoclasts.

We thank the reviewer for their suggestion. Indeed, CTSK is expressed in periosteal cells in mice and this is now acknowledged in the text. We do observe however, that the morphology of periskeletal cells is very distinctive from osteoclasts and can be identified with ease. Nevertheless, addressing the reviewer’s concern, we performed immunofluorescence for the macrophage marker F4/80 in sections from our transgenic line CTSK:eGFP at 7 dpa (Figure 2 —figure supplement 1A,B) and found most of the cells in the vicinity of the skeleton to be positive, demonstrating the immune lineage of osteoclasts in axolotls. Interestingly, we also observed some periskeletal cells to be eGFP^+^ but F4/80^-^, which might resemble the Ctsk^+^ stem cells in mice. These modifications can be found in lines 164-173.

– Line 171: Can the authors clarify if they also used in vivo imaging for these elegant transgenic assays?

Yes, the imaging was performed in vivo and this was now clarified in the text.

– I feel that the logical connection between the section prior to Line 228 and the section after Line 229 is weak. It would be important to show how the angulation is also altered in zol-treated animals. It is possible that this phenotype is completely independent of zol-treatment.

We agreed with the reviewer and we modified this section by creating a scoring method to assess the integration phenotypes observed (details in Materials and methods). We also modified the paragraph after line 229 (now 283-290) with the new results accordingly to improve the quality of our conclusions.

– Line 264: Again, it is possible that these CTSK^+^ cells include periosteal cells, which may have an important contribution to blastema formation.

Albeit this is possible, our results demonstrate that the majority of Ctsk^+^ cells are osteoclasts and have an immunological background. In addition, perioskeletal cells are morphologically different than osteoclasts.

– For Figure 5E-H, it would be important to show how the distribution of osteoclasts changes when FSF is performed.

As shown in Figure 5C, upon FSF surgery, the number of Ctsk^+^ cells (which represent in its majority osteoclasts) is strikingly reduced to almost absent. We considered that the distribution of very few, if any osteoclast would not add considerable information.

– Figure 6 is interesting but very descriptive. The authors should at least provide some validation of these predicted changes.

We share the reviewer’s concern, and we continue insisting on finding the best way to validate the GOI and explore their influence in osteoclast-mediated histolysis. We observed that there are several genes related to recruitment and/or activation of osteoclasts, in addition to MMPs, expressed in the WE. Creating Crispants or morpholinos for this battery and evaluate the phenotypes for several candidate genes would be a lengthy experiment that we look forward to tackle in a future manuscript.

– Figure 7: What is the significance of decreased blastema formation in zol-treated animals? The limb regeneration occurs anyway, only associated with a relatively minor defect in skeletal integration. The authors would need to perform EdU assays in the blastema of zol-treated animals.

With Figure 7 we aimed to challenge the current understanding of blastema and its position in the limb. Although we show that blastema size decreases in zol-treated animals, it does not have an effect in the overall limb regeneration process. As suggested by the reviewer, we performed EdU staining and show no difference in the number of proliferative cells in zol-treated animals compared to vehicle and control (Figure 7 —figure supplement 1A). The results show in Figure 7 demonstrate that the position of the blastema in the limb is not fixed to the amputation plane and that the skeleton cannot be used as a reference point to define it.

– Line 366: aggrupation (?)

Changed

– In Figure 7D, the Sox9 reporter assay is nice, but it would be important to see how these cells are disrupted in zol-treated animals, to enhance the relevance of this experiment.

The aim of Figure 7D was to show that resorption and condensation of the new skeleton occur in parallel, which is demonstrated by this image set. Nevertheless, to further support our results we included Figure 7 —figure supplement 1B, where Ctsk^+^ cells were found at the same time and in the same plane as condensating *Sox9*^+^ cells, further proving the overlapping of resorption and skeletal regeneration.

4. Discussion:– Line 411: I do not see any convincing quantitative data that support this statement, "the more resorption, the better the integration. Evidence now is somewhat anecdotal.

We agree with the reviewers and thus we have included a scoring system to quantitatively assess integration outcomes in regenerated limbs (Figure 4C, Figure 4 —figure supplement 1). By given numerical values to different aspects of the phenotype, we were able to correlate these values with the resorption rate for each skeletal element and found a correlation between resorption and better integration (as seen by lower score values).

[Editors’ note: further revisions were suggested prior to acceptance, as described below.]

Reviewer #1 (Recommendations for the authors):The manuscript has improved upon revision. However, my major concern stays the same if not get worsened. Specifically – do osteoclasts have any role in the presented story. Please see my arguments below.The author did some additional analysis in relation to this issue and said the following in their rebuttal letter "However, to our surprise, it only resulted in a decreased number and function inhibition. ". However, the data were not presented in the revised manuscript.It further rises my suspicion that osteoclasts may not be a key player involved in the observed reintegration defects.Here are additional observations and arguments making me so concerned:Macrophages are unequivocal for limb regeneration in salamanders since their ablation abrogates limb regeneration entirely (PNAS, 2013, 110 (23) 9415-9420). Zoledronic acid may directly affect macrophages (numerous references, i.e., Cellular Oncology volume 36, pages 505-514 (2013)) including polarization toward M1 (Faseb J 2019 Apr;33(4):5208-5219).Thus, the observed effect of impaired reintegration, may not be related to osteoclasts activity but to macrophages.Furthermore, the developmental stage of axolotl used in this study is 4-6 cm snout-tail (method section) at the time of limb amputation. At this stage neither ulna nor radius are yet developed bone tissue, only cartilage. Accordingly, the observed labeling with calcein reflects calcification of hypertrophic area prior to ossification starts.Thus, the observed changes in the calcified area may not necessarily reflect osteoclast activity or skeletal resorption but calcium loss due to numerous other reasons such as acidic pH of the inflammatory environment, active remodeling of extracellular matrix by MMP8/13 proteases (ref Gerber et al., Science 2018), etc.In this same paper, a single-cell RNAseq of axolotl limbs and blastema has been performed. Analysis of this dataset reveals that Ctsk is highly expressed by a large fraction of blastema cells, particularly those of mesenchymal origin, i.e., Prrx1-sorted.Thus, the observed Ctsk-GFP positive cells may not be osteoclasts but blastema-forming cells, whereas white arrowheads on histological images (Figure 5F,G) may well point toward macrophages.Thus, altogether, the observed effects of tissue reintegration may have no relation to osteoclasts at all. It puts the entire story under question.

We appreciate the interest of reviewer #1, and as we respectfully disagree with some of the statements, we would like to further explain why.

Regarding our treatments with zoledronic acid, we are unequivocally showing that there is a decrease in the resorption of the calcified matrix. In Figure 3A, B we show an inhibition in resorption. In Figure 3C, D we show that zoledronic acid does not result in a significant reduction of mRNA levels of osteoclasts markers (e.g. *Ctsk, Trap, Dcstamp*) and in our transgenic animal, we show that *Ctsk^+^* cells are still visible, while tissue resorption is clearly affected. This support our statement that although the cells are still there, the function is compromised, the cells are present and they fail to resorb the skeleton. The specificity of zoledronic acid is fairly questioned. We have used an additional experimental set up, the FSF, where we show an absence of osteoclast and lack of resorption. The reviewer’s concern is that zol could be affecting macrophages and the phenotype we see is a result of this. Indeed, the publication referred by the reviewer (PNAS, 2013, 110 (23) 9415-9420), uses clodronate to ablate macrophages, which stops regeneration. Similar to zoledronic acid, the use of clodronate to specifically ablate macrophages comes at the expense of affecting other immune cells. Transgenesis could prove useful to overcome these pitfalls, and we have included these concerns in our discussion.

The gigantic cells that we have followed in vivo, arrive at the amputation site (Figure 2D), remain in the proximity of the skeleton (Figure 5 —figure supplement 2) and engulf skeletal cells (Figure 2E), without moving into the blastema mesenchyme (Figure 5C, Figure 5 —figure supplement 2). And as a textbook definition of osteoclasts, these cells resorb skeleton, are multinucleated and are localized in proximity of skeletal structures. Additionally, our conclusions do not claim that osteoclasts are involved in the re-integration process, but rather that the active skeletal resorption mediated by osteoclasts is necessary for the following phase: tissue integration. This tissue integration is a fascinating process that probably orchestrates cell differentiation with concomitant matrix deposition, and surely deserves more detailed attention to finally understand it. When we inhibit resorption in our FSF setting, we unequivocally show the absence of *Ctsk^+^* cells, further implicating these cells in skeletal resorption.

Our results in adult animals (Figure 5 —figure supplement 1), show that resorption of ossified bone occurs to a similar extent and similar time window as in our small animals. We believe that only osteoclasts could drive this resorption of the hard, bony matrix so effectively in such a short time. This supports the evidence that osteoclasts are responsible for the regeneration-triggered skeletal resorption in axolotl. Even more striking, is the fact that re-amputation of the skeleton occurs in some cases (Figure 1F). The resulting loose piece of skeleton is then extruded through the already formed wound epidermis. Making a very localized breaking point in a hard-calcified structure and relatively fast, is very likely to be the result of osteoclast function, and not merely decalcification.

The concern that *Ctsk* is expressed in other cell types and possibly in blastema cells is addressed in this work, by providing evidence of the morphology and location of these cells (see above). The reviewer mentions the publication of our colleagues (Gerber et al.) and as also correctly states that the RNAseq was only performed using *Prrx1^+^* cells. With that dataset, it is impossible to have a point of comparison of transcript counts with cells that do not express *Prrx1*, such as immune cells and other blastema cells. On the other hand, Tsai et al. 2020, performed a RNAseq from three sorted populations of cells during regeneration: blastema cells, mature cells and epithelial cells. In that work, *Ctsk* is highly expressed in mature cells and not in blastema cells (shown in our Figure 6 and Tsai et al. 2020). Moreover, our work provides images of the *Ctsk* transgenic that clearly shows the location of the positive cells. Particularly, in Figure 5 —figure supplement 2, the position of *Ctsk^+^* cells is almost entirely observed surrounding the skeletal elements, while blastemal cells (negative for *Ctsk*) are observed distal to them. Finally, in Figure 7 —figure supplement 1, no *Ctsk^+^/Sox9^+^* cell is observed, suggesting that no chondroprogenitor is expressing *Ctsk* during the early phases of cartilage condensation.

This fast resorptive process could imply that differentiation of osteoclasts is also occurring simultaneously, which could involve fusion of cells of different origin occurs, including macrophages. Additionally, as we observed mononucleated *Ctsk^+^* cells*,* it would be interesting to know if they are posed to rapidly behave as skeletal resorbing cells. It is clear to us that more studies to unravel the axolotl immune system are required, such as determining molecular signatures for different populations and also untangling how those populations are participating during regeneration. Moreover, the differentiation process of myeloid cells to contribute to the osteoclast population, as well as the contribution of circulating and resident macrophages will need to be further explored. These challenges were already included in our “Future Perspective” section in the discussion, but in order to address the Reviewer’s concerns, we have included more details regarding the pitfalls from our work.

We would like to emphasize that it is important to understand how axolotl regeneration diverts from what we know in mammalian systems. For example, we have recently published a developmental characterization of the axolotl limb skeleton, and surprisingly found a population of cells expressing Osteocalcin and *SOX9* in homeostatic conditions. The presence of a hybrid cell type, expressing chondrocyte and osteoblast genes, occurs in zebrafish and mouse but only during an injury response, further underscoring species-specific differences.

Reviewer #3 (Recommendations for the authors):The authors revised the manuscript in response to the comments from three reviewers during the previous round of peer review. The authors performed additional experiments to support the conclusion further. Below, I am going to evaluate the authors' responses to my comments.1. The authors did not clarify that they used zoledronic acid to inhibit bone resorption in the abstract, as requested. It is extremely important to specify in the abstract that they used zol, but not other reagents, to inhibit resorption. This is an essential piece of information to support the overall integrity of the work.

We apologize for the mistake. This is now clearly stated in the abstract.